# Learning Exposure Mapping Functions for Inferring Heterogeneous Peer Effects

**Shishir Adhikari**[*]  **Sourav Medya**    **Elena Zheleva**
University of Illinois Chicago, Chicago, IL, USA
{sadhik9,medya,ezheleva}@uic.edu

## Abstract

Peer effect refers to the difference in counterfactual outcomes for a unit resulting from different levels of peer exposure, the extent to which the unit is exposed to the treatments, actions, or behaviors of its peers. Peer exposure is typically captured through an explicitly defined exposure mapping function that aggregates peer treatments and outputs peer exposure. Exposure mapping functions range from simple functions like the number or fraction of treated friends to more sophisticated functions that allow for different peers to exert different degrees of influence. However, the true function is rarely known in practice and when the function is misspecified, this leads to biased causal effect estimation. To address this problem, the focus of our work is to move away from the need to explicitly define an exposure mapping function and instead introduce a framework that allows learning this function automatically. We develop EGONETGNN, a graph neural network (GNN), for heterogeneous peer effect estimation that automatically learns the appropriate exposure mapping function and allows for complex peer exposure mechanisms that involve not only peer treatments but also attributes of the local neighborhood, including node, edge, and structural attributes. We theoretically and empirically show that GNN models that use peer exposure based on the number or fraction of treated peers or learn peer exposure naively face difficulty accounting for such influence mechanisms. Our evaluation on synthetic and semi-synthetic network data shows that our method is more robust to different unknown underlying influence mechanisms when compared to state-of-the-art baselines.

## 1 Introduction

In networked environments, the outcome of a unit can be influenced by the treatments or outcomes of other units, a phenomenon known as interference. For example, in a contact network, the smoking habits of peers may affect an individual's respiratory health, and in a social network the political affiliations of peers may influence one's stance on a policy issue like immigration. Peer effects capture this influence by comparing an individual's outcomes under different peer network conditions (e.g., having no smoker peers versus some smoker peers, or observed peer political affiliations versus counterfactual, flipped affiliations). Peer effect estimation is important for policy-making and targeted intervention design in many domains, including healthcare (Barkley et al., 2020), online advertisement (Nabi et al., 2022), and education (Patacchini et al., 2017).

Peer network conditions are typically captured through an *explicitly defined* exposure mapping function (Aronow and Samii, 2017) that summarizes the peer treatments and peer network and outputs peer exposure, which is the equivalent to a composite peer treatment value. The peer effect is defined as the difference in outcomes under two distinct levels of peer exposure. Different peer exposure mapping functions capture different possible underlying influence mechanisms. Typically, domain experts define exposure mapping functions appropriate to the causal question and domain of interest. The advantage of exposure mapping functions is that they reduce the high dimensionality of peer network attributes and that they are invariant to irrelevant contexts (e.g., permutation of peers).

Figure 1 presents examples of prominent exposure mapping functions and the resulting peer exposure values for a toy peer network. The first graph shows Gaby's peer network along with the observed

---

[*]Currently affiliated to Icahn School of Medicine at Mount Sinai, NY, USA.

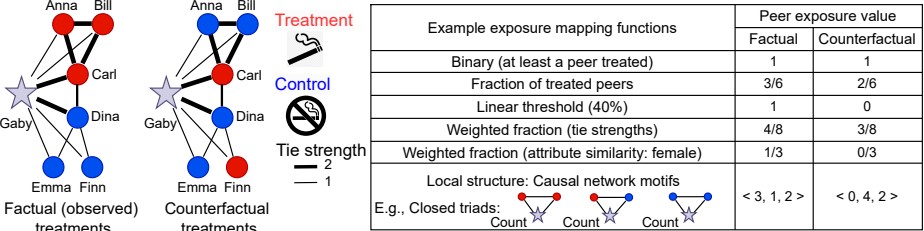

Figure 1: Illustration of different possible peer exposure representations for a node (Gaby) in a toy peer network. Red nodes represent peers in the treatment group, and blue nodes represent peers in the control group. Gray star node represents the node that has a fixed treatment.

(i.e., factual) treatments for Gaby's peers. The second graph shows hypothetical (i.e., counterfactual) treatments for the peers. The peers in the treatment group (e.g., smokers) and control group (e.g., non-smokers) are depicted as red and blue nodes, respectively. The edge weights capture the tie strengths in the network. Binary peer exposure mapping is the simplest and it summarizes peer treatments to 0 or 1, e.g., whether any peers have been treated (Bargagli-Stoffi et al., 2025) or whether the weighted treatment of peers has reached a linear threshold (Tran and Zheleva, 2022). Some exposure mapping functions assume that all peers influence equally (e.g., fraction of treated peers (Hudgens and Halloran, 2008; Jiang and Sun, 2022)), while others consider that different peers can exert different degrees of influence (e.g., weighted fraction (Forastiere et al., 2021) or sum (Zhao et al., 2024) of treated peers). Peer exposure has also been modeled with counts of different causal network motifs, i.e., recurrent subgraphs in a unit's peer network with treatment assignments as attributes (Yuan et al., 2021). We discuss the related work in more detail in the Appendix A.2.

A key challenge in peer effect estimation is that the true exposure mapping function is rarely known in practice and when the function is misspecified, this leads to biased causal effect estimation. The focus of this paper is to move away from the need to explicitly define an exposure mapping function and instead learn this function automatically from data. This has the advantage of reducing subjectivity and allowing for automated representation of peer exposure under unknown and complex peer influence mechanisms. More specifically, we study the problem of exposure mapping function learning in the context of heterogeneous peer effect estimation. Heterogeneous peer effects (HPE) denote variation in peer effects across individuals that may originate from personal attributes or from characteristics of their peer networks. For example, while having a friend who smokes may have a negative effect on health for some people, it may make no difference for others.

We propose EGONETGNN, a novel graph neural network (GNN) architecture, that automatically learns a relevant exposure mapping function under appropriate identifiability assumptions. EGONET-GNN allows for complex peer influence mechanisms that, in addition to peer treatments, can involve the local neighborhood structure, node, and edge attributes. Our work builds upon the success of utilizing neural networks (NNs) (Shalit et al., 2017; Im et al., 2021; Shi et al., 2019) and graph neural networks (GNNs) (Jiang and Sun, 2022; Cai et al., 2023; Chen et al., 2024; Khatami et al., 2024) for end-to-end learning of counterfactual outcome models or causal effect estimators. Few studies have utilized GNNs to learn the exposure mapping function (Mao et al., 2025; Wu et al., 2025) or to derive peer exposure embedding by aggregating feature embeddings and peer treatments (Adhikari and Zheleva, 2025; Zhao et al., 2024). However, these works use off-the-shelf GNNs like GCN (Kipf and Welling, 2016) or GIN (Xu et al., 2018) and prior work (Chen et al., 2020) has shown such architectures lack expressiveness for counting subgraphs with cycles and for capturing mechanisms involving local neighborhood structure. Yet counts of such subgraphs, like causal network motifs, are rich features for capturing local structural contexts (Yuan et al., 2021), but they are expensive to compute, inflexible, and may not capture every local structural context (e.g., edge weights).

EGONETGNN captures exposure mapping functions from previous works, including relevant causal network motifs and scaling to higher-order motifs. To add robustness to the downstream peer effect estimation task, EGONETGNN is designed to learn the exposure mapping function to produce representation that is *expressive* to differentiate between different peer exposure conditions and *invariant* to irrelevant contexts. Moreover, EGONETGNN is designed to promote bounded representation with substantial coverage of possible peer exposure values. Figure 2 shows an overview of EGONETGNN. While most peer effect estimation frameworks contain a feature mapping and a counterfactual out-

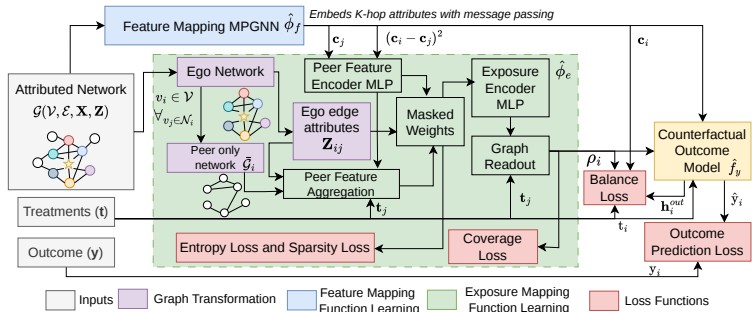

Figure 2: An overview of the proposed EGONETGNN model to learn exposure mapping function for peer effect estimation. EGONETGNN extracts ego networks, for each node $v_i$, with peer treatments along with feature embedding and ego edge attributes as peer features. Then, node-level aggregations are performed to capture local neighborhood contexts. These contexts are passed through a masked weight layer and encoded by an multi-layer perceptron (MLP) to learn relevant influence mechanisms and summarized with graph-level aggregation. The learned peer exposure embeddings ($\boldsymbol{\rho}_i$), along with the feature embeddings ($\mathbf{c}_i$), and treatment ($t_i$) are passed to a counterfactual outcome model that is used to infer peer effects. The graph transformation ensures expressiveness, while balance, coverage, entropy, and sparsity losses promote the robustness of the peer exposure representation.

come model component, the novel additional component in ours is the custom-designed exposure mapping function learning (marked in green in the figure). We design this component to excel in counting attributed subgraphs, such as causal network motifs, enhancing its expressiveness to capture unknown underlying peer exposure mechanisms. We theoretically and empirically show that, unlike EGONETGNN, existing GNN-based approaches that solely rely on homogeneous peer exposure or learn heterogeneous peer exposure naively lack expressiveness in capturing heterogeneous peer influence mechanisms based on local neighborhood structure.

## 2 CAUSAL INFERENCE PROBLEM SETUP

**Notations**. We represent the network as an undirected graph $\mathcal{G} = (\mathcal{V}, \mathcal{E})$ with a set of $n = |\mathcal{V}|$ nodes, a set of edges $\mathcal{E}$, node attributes $\mathbf{X}$, and edge attributes $\mathbf{Z}$. Let $\mathbf{t} = < t_1, ..., t_i, ..., t_n >$ be a random variable comprising the treatment variables $t_i$ for each node $v_i \in \mathcal{V}$ in the network and $y_i$ be a random variable for $v_i$'s outcome. Let $\boldsymbol{\pi} = < \pi_1, ..., \pi_i, ..., \pi_n >$ be an assignment to $\mathbf{t}$ with $\pi_i \in \{0, 1\}$ assigned to $t_i$. Let $\mathbf{t}_{-i} = \mathbf{t} \setminus t_i$ and $\boldsymbol{\pi}_{-i} = \boldsymbol{\pi} \setminus \pi_i$ denote random variable and its value for treatment assignment to other units except $v_i$, which we refer to as peer treatments for convenience.

Peer exposure reflects how much a unit is exposed to peer treatments and is defined as follows.

**Definition 1** (Peer exposure and exposure mapping function). Peer exposure for unit $v_i$ is defined as $\boldsymbol{\rho}_i \in [0, 1]^d = \phi_e(\boldsymbol{\pi}_{-i}, \mathcal{G}, \mathbf{X}, \mathbf{Z})$, where $\phi_e$ is the *exposure mapping function* that maps high-dimensional contexts $\{\boldsymbol{\pi}_{-i}, \mathcal{G}, \mathbf{X}, \mathbf{Z}\}$ to a $d$-dimensional peer exposure representation bounded between 0 and 1 such that $y_i(t_i = \pi_i, \mathbf{t}_{-i} = \boldsymbol{\pi}_{-i}) | \{\mathcal{G}, \mathbf{X}, \mathbf{Z}\} = y_i(t_i = \pi_i, \mathbf{p}_i = \boldsymbol{\rho}_i) | \{\mathcal{G}, \mathbf{X}, \mathbf{Z}\}$.

Definition 1 maps peer treatments $\mathbf{t}_{-i} = \boldsymbol{\pi}_{-i}$ and peer network contexts $\{\mathcal{G}, \mathbf{X}, \mathbf{Z}\}$ to peer exposure $\mathbf{p}_i = \boldsymbol{\rho}_i$ in terms of equivalence of counterfactual outcomes $y_i(t_i = \pi_i, \mathbf{t}_{-i} = \boldsymbol{\pi}_{-i})$ and $y_i(t_i = \pi_i, \mathbf{p}_i = \boldsymbol{\rho}_i)$. Here, $y_i(t_i = \pi_i, \mathbf{p}_i = \boldsymbol{\rho}_i)$, captures that, in interference settings, the counterfactual outcome of a unit $v_i$ is influenced by both unit's treatment $t_i = \pi_i$ and peer exposure $\mathbf{p}_i = \boldsymbol{\rho}_i$. Note that the exposure mapping function could map different contexts to the same peer exposure.

Peer effect refers to the difference in counterfactual outcomes for different values of peer exposure. Heterogeneous peer effects (HPE) refers to different units having different peer effects dependent on their contexts. For any given unit $v_i$, its heterogeneous peer effect is described through its context, i.e., for peer exposures $\mathbf{p}_i = \boldsymbol{\rho}_i$ versus $\mathbf{p}_i = \boldsymbol{\rho}'_i$ and unit's treatment $t_i = \pi_i$ conditioned on the unit's contexts $\mathbf{c}_i$, it is defined as:

$$\delta_i(\boldsymbol{\rho}_i, \boldsymbol{\rho}'_i) = \mathbb{E}[y_i(t_i = \pi_i, \mathbf{p}_i = \boldsymbol{\rho}_i) | \mathbf{c}_i] - \mathbb{E}[y_i(t_i = \pi_i, \mathbf{p}_i = \boldsymbol{\rho}'_i) | \mathbf{c}_i], \tag{1}$$

where expectation is over units with similar contexts $\mathbf{c}_i$, referred to as *effect modifiers* (e.g., unit's degree or node attributes), defined by a feature mapping function of contexts $\{\mathcal{G}, \mathbf{X}, \mathbf{Z}\}$ from $v_i$'s perspective, i.e., $\mathbf{c}_i = \phi_f(v_i, \mathcal{G}, \mathbf{X}, \mathbf{Z})$. Substituting peer exposures $\boldsymbol{\rho}_i$ and $\boldsymbol{\rho}'_i$ with corresponding

exposure mapping functions for two peer treatment assignments $\boldsymbol{\pi}_{-i}$ versus $\boldsymbol{\pi}'_{-i}$ in Eq. 1, we get:

$$\delta_i(\boldsymbol{\pi}_{-i}, \boldsymbol{\pi}'_{-i}) = \mathbb{E}[y_i(t_i = \pi_i, \mathbf{p}_i = \phi_e(\boldsymbol{\pi}_{-i}, \mathcal{G}, \mathbf{X}, \mathbf{Z}))|\mathbf{c}_i] - \mathbb{E}[y_i(t_i = \pi_i, \mathbf{p}_i = \phi_e(\boldsymbol{\pi}'_{-i}, \mathcal{G}, \mathbf{X}, \mathbf{Z}))|\mathbf{c}_i]. \quad (2)$$

**Causal identification**. Now, we discuss the identification of peer effects that involves expressing counterfactual outcomes in terms of observational and/or interventional distributions. First, we make two commonly adopted assumptions in network interference settings.

**Assumption 1** (Pre-treatment network)**.** The network $\mathcal{G}$ along with node attributes $\mathbf{X}$ and edge attributes $\mathbf{Z}$ are measured before treatment assignments $\mathbf{t} = \boldsymbol{\pi}$ and treatments are not mutable.

**Assumption 2** (Neighborhood Interference)**.** The counterfactual outcome of a unit depends only on its immediate neighborhood treatments, i.e., $y_i(t_i = \pi_i, \mathbf{t}_{-i} = \boldsymbol{\pi}_{-i})|\mathbf{c}_i = y_i(t_i = \pi_i, \mathbf{t}^{\mathcal{N}_i}_{-i} = \boldsymbol{\pi}^{\mathcal{N}_i}_{-i})|\mathbf{c}_i = y_i(t_i = \pi_i, \mathbf{p}_i = \phi_e(\boldsymbol{\pi}^{\mathcal{N}_i}_{-i}, \mathcal{G}, \mathbf{X}, \mathbf{Z}))|\mathbf{c}_i$, where $\mathcal{N}_i = \{j : (v_i, v_j) \in \mathcal{E}\}$, $\mathbf{t}^{\mathcal{N}_i}_{-i} = \mathbf{t}_{-i} \cap \{t_j : j \in \mathcal{N}_i\}$, and $\boldsymbol{\pi}^{\mathcal{N}_i}_{-i} = \boldsymbol{\pi}_{-i} \cap \{\pi_j : j \in \mathcal{N}_i\}$ denote neighborhood set, treatments, and assignments, respectively.

Assumption 1 is a general assumption in experimental and observational studies, and Assumption 2 is a common simplifying assumption that presumes network influence is mediated by immediate neighbors but our work could be extended to consider interference from multiple-hop neighborhoods. For ease of exposition, we drop the superscript $\mathcal{N}_i$ in neighborhood treatments and assignments.

For causal identification, we assume unconfoundedness, similar to previous work (Ma et al., 2022; Wu et al., 2025):

**Assumption 3** (Unconfoundedness)**.** For all unit treatment $\pi_i \in \{0, 1\}$ and peer treatment assignments $\boldsymbol{\pi}_{-i} \in \{0, 1\}^{n-1}$, there exists a feature mapping function $\phi_f \in \Phi_f$ and an exposure mapping function $\phi_e \in \Phi_e$ such that the counterfactual outcome is independent of unit treatment and peer exposure conditions given the context $\mathbf{c}_i = \phi_f(v_i, \mathcal{G}, \mathbf{X}, \mathbf{Z})$, i.e., $y_i(t_i = \pi_i, \mathbf{p}_i = \phi_e(\boldsymbol{\pi}_{-i}, \mathcal{G}, \mathbf{X}, \mathbf{Z})) \perp \{t_i, \mathbf{p}_i\}|\mathbf{c}_i$.

Assumption 3 implies that the observed network context is sufficient for controlling for confounding, and there are functions able to represent it compactly. Under this assumption, it is still possible to learn a feature mapping and exposure mapping functions that do not approximate the true functions which leads to a misspecification error. Therefore, it is important to learn an expressive function (e.g., a GNN) that is able to capture a wide range of possible functions. We also assume the standard *consistency* (Assumption 4) and *positivity* (Assumption 5), described in more detail in Appendix A.3. Next, we present the causal identification conditions and formally define the problem of exposure mapping function learning in the context of peer effect estimation.

**Proposition 1.** With Assumptions 1-5, the HPE $\delta_i$ in Eq. A.3 can be estimated from experimental or observational data as

$$\delta_i(\boldsymbol{\pi}_{-i}, \boldsymbol{\pi}'_{-i}) = \mathbb{E}[y_i|t_i = \pi_i, \mathbf{p}_i = \phi_e(\boldsymbol{\pi}_{-i}, \mathcal{G}, \mathbf{X}, \mathbf{Z}), \mathbf{c}_i = \phi_f(v_i, \mathcal{G}, \mathbf{X}, \mathbf{Z})]$$
$$- \mathbb{E}[y_i|t_i = \pi_i, \mathbf{p}_i = \phi_e(\boldsymbol{\pi}'_{-i}, \mathcal{G}, \mathbf{X}, \mathbf{Z}), \mathbf{c}_i = \phi_f(v_i, \mathcal{G}, \mathbf{X}, \mathbf{Z})].$$

The proof presented in Appendix A.3 stems from consistency and unconfoundedness assumptions.

**Problem 1** (Exposure mapping function learning)**.** Given network contexts $\{\mathcal{G}, \mathbf{X}, \mathbf{Z}\}$, treatments $\mathbf{t}$, and outcomes $\mathbf{y}$ of $n$ units, estimate the feature and exposure mapping functions $\hat{\phi}_f$ and $\hat{\phi}_e$ along with counterfactual outcome model $\hat{f}_y$ such that mean squared error between true heterogeneous peer effect (HPE) $\delta_i$ and estimated HPE $\hat{\delta}_i$, i.e., $\frac{1}{n}\sum_{i=1}^{n}(\delta_i - \hat{\delta}_i)^2$, is minimized, where $\hat{\delta}_i = \hat{f}_y(\pi_i, \hat{\boldsymbol{\rho}}_i, \hat{\mathbf{c}}_i) - \hat{f}_y(\pi_i, \hat{\boldsymbol{\rho}}'_i, \hat{\mathbf{c}}_i)$ with $\hat{\boldsymbol{\rho}}_i = \hat{\phi}_e(\boldsymbol{\pi}_{-i}, \mathcal{G}, \mathbf{X}, \mathbf{Z})$, $\hat{\boldsymbol{\rho}}'_i = \hat{\phi}_e(\boldsymbol{\pi}'_{-i}, \mathcal{G}, \mathbf{X}, \mathbf{Z})$, and $\hat{\mathbf{c}}_i = \hat{\phi}_f(v_i, \mathcal{G}, \mathbf{X}, \mathbf{Z})$.

The true HPE is unknown, but due Proposition 1, the factual outcomes can be utilized to jointly estimate $\hat{\phi}_f$, $\hat{\phi}_e$, and $\hat{f}_{Y_i}$ as discussed in the next section.

## 3 EGONETGNN: LEARNING EXPOSURE MAPPING FUNCTION WITH GNNs

Figure 2 shows an overview of the proposed EGONETGNN model to simultaneously learn exposure mapping function $\hat{\phi}_e$, feature mapping function $\hat{\phi}_f$, and counterfactual outcome model $\hat{f}_y$ for peer effect estimation. We aim to learn exposure mapping function $\hat{\phi}_e$ with three key properties: 1) expressiveness, 2) invariance, and 3) bounded and balanced representation. The expressiveness property ensures the peer exposure representation $\boldsymbol{\rho}_i$ returned by the function $\hat{\phi}_e$ is unique for

different relevant contexts, while the invariance property assures the representation $\rho_i$ does not vary due to irrelevant contexts. For example, in Figure 1, if the underlying peer influence depends on clustering coefficients among treated, the function $\hat{\phi}_e$ is expressive if it can capture the first closed triad substructure. The standard message passing GNNs (e.g., GCN, GIN, etc) cannot capture essential causal network motifs like closed triads (i.e., triangular motifs) (Chen et al., 2020). The graph transformation and automated exposure mapping function learning in our EGONETGNN model are designed to ensure that the peer exposure representation is at least as expressive as or superior to the approach of feature extraction by counting causal network motifs. In the above example, the function $\hat{\phi}_e$ is invariant to irrelevant contexts if the difference in other features like node attributes and edge weights do not change the learned representation $\rho_i$. To satisfy the property of bounded representation, the learned representation $\rho_i$ should be bounded, e.g., between 0 and 1, to reflect no exposure and maximum exposure. Moreover, the representation $\rho_i$ should have a substantial coverage, which means it should be distributed across the possible range of exposure. Next, we describe our feature mapping, exposure mapping, and counterfactual outcome model in detail.

## 3.1 ARCHITECTURE OF EGONETGNN

EGONETGNN first maps the attributed network to feature embedding using a MPGNN and extracts ego networks for each node $v_i$, incorporating peer treatments, node features, and edge attributes. It performs node-level aggregation to capture local context, which is processed through a masked weight layer and an MLP followed by graph-level aggregation to learn peer exposure representation.

**Feature mapping MPGNN**. The feature mapping module aims to capture contexts that are potentially confounders or effect modifiers. Let $\Theta$ denote a multi-layer perceptron (MLP) and $||$ denote a concatenation operator. The feature embedding $c_i$ is obtained for $l$-th layer as:

$$c_i = \Theta_0(\boldsymbol{X}_i)||\boldsymbol{h}_i^l \text{ and } \boldsymbol{h}_i^l = \boldsymbol{h}_i^{l-1} + \sum_{j \in \mathcal{N}_i} \Theta_l \boldsymbol{h}_j^{l-1}, \tag{3}$$

where $\boldsymbol{h}_j^0 = \boldsymbol{X}_j||\boldsymbol{Z}_{ij}$, and $\boldsymbol{h}_i^0 = 0$ are initial conditions and $\mathcal{N}_i$ denote neighbors of node $v_i$. This MPGNN architecture incorporates edge attributes $\boldsymbol{Z}_{ij}$ while disentangling the hidden representation of the unit's own attributes $\Theta_0(\boldsymbol{X}_i)$ from that of aggregated peer and edge attributes $\boldsymbol{h}_i^l$.

**Ego network construction**. To learn an exposure mapping function that is as least as expressive as or superior to the approach of feature extraction by counting network motifs, we transform the node regression task to graph regression by extracting ego networks for each unit. In the extracted network, the triangle structures involving an ego node are transformed as dyads, which mitigates the limitation of GNNs to capture closed triad motifs. The ego network $\bar{\mathcal{G}}_i(\bar{\mathcal{V}}_i, \bar{\mathcal{E}}_i)$ is extracted from $\mathcal{G}(\mathcal{V}, \mathcal{E})$ for each node $v_i$ such that node set $\bar{\mathcal{V}}_i$ consists neighbors of $v_i$, i.e., $\bar{\mathcal{V}}_i = \{v_j : e_{ij} \in \mathcal{E} \wedge v_j \in \mathcal{V}\}$ and edge set $\bar{\mathcal{E}}_i$ consists edges between neighbors of $v_i$, i.e., $\bar{\mathcal{E}}_i = \{e_{jk} : e_{jk} \in \mathcal{E} \wedge v_j \in \bar{\mathcal{V}}_i \wedge v_k \in \bar{\mathcal{V}}_i\}$.

**Peer feature encoder and aggregation**. Peer feature encoder module takes relevant peer feature embeddings and the distance between ego and peer feature embeddings, i.e., $c_{ij} = \Theta_{feat}(c_j||(c_i - c_j)^2)$, to capture peer influence mechanisms involving peer attributes and feature similarity between ego and peers. Then, we transform an ego $v_i$'s edge attributes $\boldsymbol{Z}_{ij}$ to node attributes, i.e., $\bar{\boldsymbol{X}}_j = \boldsymbol{Z}_{ij}$, in the ego network $\bar{\mathcal{G}}_i(\bar{\mathcal{V}}_i, \bar{\mathcal{E}}_i)$ because the ego $v_i$ itself is not present in the ego network. The node aggregation for each node $v_j$ in the ego network $\bar{\mathcal{G}}_i$ considers neighbors' node attributes $\bar{\boldsymbol{X}}_k$, feature encoding $c_{ik}$, edge attributes $\boldsymbol{Z}_{jk}$, and neighbor treatments $t_k$, and is defined for $l^{th}$ layer as follows:

$$\boldsymbol{h}_j^l = \boldsymbol{h}_j^{l-1} + \sum_{k \in \mathcal{N}_j} \boldsymbol{h}_k^{l-1}, \text{ with } \boldsymbol{h}_k^0 = t_k||\bar{\boldsymbol{X}}_k||c_{ik}||\boldsymbol{Z}_{jk} \text{ and } \boldsymbol{h}_j^0 = 0. \tag{4}$$

**Masked weights and exposure encoder**. Masked weights promotes representation that is invariant to irrelevant contexts and feeds the concatenation of node attributes and hidden state after $L$ layers of node aggregation, i.e., $\boldsymbol{h}_j^{agg} = \bar{\boldsymbol{X}}_j||c_{ij}||\boldsymbol{h}_j^L$, through a *masked fully connected* layer as follows:

$$\boldsymbol{h}_j^{mask} = ReLU((\sigma(\mathbf{W}_{mask}) \odot \mathbf{W}_{agg})\boldsymbol{h}_j^{agg} + \mathbf{b}_{agg}), \tag{5}$$

where $ReLU$ and $\sigma$ are a rectified linear unit and sigmoid activation functions, $\odot$ indicates element-wise product, $\mathbf{W}_{mask}$ and $\mathbf{W}_{agg}$ are the weight matrices, and $\mathbf{b}_{agg}$ is the bias vector. The masked hidden representation $\boldsymbol{h}_j^{mask}$ is passed into an exposure encoder MLP to extract a low dimensional embedding. The goal of this module is to capture complex mechanisms based on the local neighborhood and reduce dimensionality. Formally, the output embedding $\boldsymbol{h}_j^{exp}$ is obtained as follows:

$$\boldsymbol{h}_j^{exp} = ReLU(\Theta_{exp}(ln(ReLU(\Theta_{enc}(\boldsymbol{h}_j^{mask})) + 1))), \tag{6}$$

$\Theta_{enc}$ and $\Theta_{exp}$ are two MLPs and $ln$ denotes log transformation that offers the benefit of rescaling features with large values that are significant in scale-free networks (e.g., online social networks) and introduces inductive bias to capture mechanisms involving ratios.

**Graph readout**. Finally, the peer exposure embedding $\boldsymbol{\rho}_i$ for node $v_i$ is obtained by aggregating the representation $\boldsymbol{h}_j^{exp}$ for all $v_j \in \bar{\mathcal{V}}_i$ on the entire ego network $\bar{\mathcal{G}}_i(\bar{\mathcal{V}}_i, \bar{\mathcal{E}}_i)$ as $\boldsymbol{\rho}_i = \sum_j (t_j \times \boldsymbol{h}_j^{exp})/\sum_j \boldsymbol{h}_j^{exp} || 1 - e^{-\sum_j (t_j \times \boldsymbol{h}_j^{exp})}$. We consider two aggregations such that the peer exposure embedding is bounded between 0 and 1, with 0 being the case of no peer exposure. The first aggregation captures proportion similar to the fraction of treated peers, but we weight each peer by $\boldsymbol{h}_j^{exp}/\sum_j \boldsymbol{h}_j^{exp}$ learned by the preceding layer. The second aggregation captures scale and is analogous to the number of treated peers, except that each peer is weighted by $\boldsymbol{h}_j^{exp}$.

### 3.2 END-TO-END LEARNING OF EGONETGNN

The resulting peer exposure embeddings ($\boldsymbol{\rho}_i$) and the feature embeddings ($\boldsymbol{c}_i$) from the above module along with unit treatment ($\pi_i$) are passed to a counterfactual outcome model $f_y(\text{t}_i = \pi_i, \mathbf{p}_i = \boldsymbol{\rho}_i, \mathbf{c}_i = \boldsymbol{c}_i)$ to obtain conditional counterfactual outcome $\mathbb{E}[y_i(\text{t}_i = \pi_i, \mathbf{p}_i = \boldsymbol{\rho}_i)|\mathbf{c}_i = \boldsymbol{c}_i] = \mathbb{E}[y_i | \text{t}_i = \pi_i, \mathbf{p}_i = \boldsymbol{\rho}_i, \mathbf{c}_i = \boldsymbol{c}_i]$ (Eq. **??**). We adapt the Treatment Agnostic Representation Network (TARNet) and Counterfactual Regression (CFR) models (Shalit et al., 2017) as the counterfactual outcome model $\hat{f}_y$. The TARNet architecture consists of a single embedding MLP and two prediction heads to estimate counterfactual outcomes with unit treatment $\text{t}_i = 1$ and $\text{t}_i = 0$, i.e.,

$$\boldsymbol{h}_i^{emb} = \Theta_{emb}(\boldsymbol{c}_i)||\boldsymbol{\rho}_i, \quad \hat{\text{y}}_i(0) = \Theta_{y_0}(\boldsymbol{h}_i^{emb}), \quad \hat{\text{y}}_i(1) = \Theta_{y_1}(\boldsymbol{h}_i^{emb}). \quad (7)$$

Our CFR$^+$ architecture is similar except for an autoencoder to produce the embeddings, i.e., $\boldsymbol{h}_i^{emb} = \Theta_{emb}(\boldsymbol{c}_i||\boldsymbol{\rho}_i)$ and $\boldsymbol{h}_i^{out} = \Theta_{dec}(\boldsymbol{h}_i^{emb})$. Note that, unlike the original CFR, our CFR$^+$ utilizes an autoencoder because it, along with reconstruction loss, helps mitigate the potential loss in expressiveness while balancing representations across treatment groups. The CFR$^+$ or TARNet model $\hat{f}_y(\pi_i, \boldsymbol{\rho}_i, \boldsymbol{c}_i)$ predicts outcome $\hat{y}_i = \hat{y}_i(1)$ if $\pi_i = 1$ and $\hat{y}_i = \hat{y}_i(0)$ if $\pi_i = 0$. The unit-level factual prediction loss $\mathcal{L}_{y_i}$ is defined as

$$\mathcal{L}_{y_i} = loss(y_i, \hat{f}_y(\text{t}_i = \pi_i, \hat{\mathbf{p}}_i = \hat{\phi}_e(\boldsymbol{\pi}_{-i}, \mathcal{G}, \boldsymbol{X}, \boldsymbol{Z}; \Theta_e), \hat{\boldsymbol{c}}_i = \hat{\phi}_f(v_i, \mathcal{G}, \boldsymbol{X}, \boldsymbol{Z}; \Theta_f); \Theta_y)), \quad (8)$$

where $loss$ is an appropriate loss function (e.g., square error loss) based on data type of the outcome and $\boldsymbol{\Theta} = \{\Theta_e, \Theta_f, \Theta_y\}$ are learning parameters to be optimized for exposure mapping function $\hat{\phi}_e$, feature mapping function $\hat{\phi}_f$, and counterfactual outcome model $\hat{f}_y$, respectively.

**Balance loss**. The CFR$^+$ architecture uses autoencoder reconstruction loss and the Integral Probability Metric (IPM) (Shalit et al., 2017) measure of distance between treatment and control groups using Wasserstein (Cuturi and Doucet, 2014; Arjovsky et al., 2017), jointly referred to as balance loss, i.e.,

$$\mathcal{L}_{bal} = \mathbf{1}_{\lambda_{bal}>0} \times \frac{1}{n} \sum_i (\boldsymbol{h}_i^{out} - \boldsymbol{c}_i||\boldsymbol{\rho}_i)^2 + \lambda_{bal} \times IPM(\{\boldsymbol{h}_i^{emb} : t_i = 1\}, \{\boldsymbol{h}_i^{emb} : t_i = 0\}), \quad (9)$$

where $\lambda_{bal} \geq 0$ is a hyperparameter and $IPM(.)$ balances the distribution $\mathbb{P}(\mathbf{c}, \mathbf{p}|\text{t} = 0)$ and $\mathbb{P}(\mathbf{c}, \mathbf{p}|\text{t} = 0)$, where $\mathbb{P}(\mathbf{c}, \mathbf{p}|\text{t})$ is equivalent to $\mathbb{P}(\mathbf{p}|\text{t})\mathbb{P}(\mathbf{c}|\mathbf{p}, \text{t})$. Intuitively, $\mathcal{L}_{bal}$ balances peer exposure distribution $\mathbf{p}$ between treatment groups and covariate distribution $\mathbf{c}$ across peer exposure conditions and treatment groups while maintaining expressiveness due to the autoencoder component. Although the ideal objective would be to balance representation across any exposure conditions, i.e., $\mathbb{P}(\mathbf{c}|\mathbf{p}, \text{t}) \approx \mathbb{P}(\mathbf{c}|\mathbf{p}', \text{t}')$, our balancing technique is still a computation-friendly and useful heuristic. We discuss more on the practical infeasibility of implementing the ideal objective in the Appendix.

For the end-to-end learning of $\hat{\phi}_e$, $\hat{\phi}_f$, and $\hat{f}_{Y_i}$, we introduce three custom loss functions designed for EGONETGNN: coverage loss, sparsity loss, and entropy loss. These custom loss functions serve as priors to make the learned exposure mapping function stable and reliable.

**Coverage loss**. We use a prior that encourages the bounded peer exposure embedding to have substantial coverage. This loss function checks how far the learned peer embedding distribution is from a continuous uniform distribution between 0 and 1, i.e., $L_{cov} = (mean(\boldsymbol{\rho}) - 0.5)^2 + (var(\boldsymbol{\rho}) - \frac{1}{12})^2 + (range(\boldsymbol{\rho}) - 1)^2$. Here, we consider mean squared error of mean, variance, and range of learned embedding $\boldsymbol{\rho}$ against corresponding value of the uniform distribution.

**Entropy loss and sparsity loss**. Entropy loss encourages mask weights, i.e., $p := \sigma(\mathbf{W}_{mask})$ to take values toward 0 or 1 and sparsity loss pushes for a few weights with high values. Formally, we define entropy loss and sparsity loss as $\mathcal{L}_{ent} = mean(-p log(p) - (1-p) log(1-p))$ and $\mathcal{L}_{sp} = mean(p)$.

**Overall loss**. We obtain the overall loss function $\mathcal{L}$ as

$$\mathcal{L} = \frac{1}{n} \sum_i \mathcal{L}_{y_i} + \mathcal{L}_{bal} + \lambda_{cov} \times \mathcal{L}_{cov} + \lambda_{ent} \times \mathcal{L}_{ent} + \lambda_{sp} \times \mathcal{L}_{sp} + \lambda_{L1} \times ||\mathbf{\Theta}_{gnn}||_1, \quad (10)$$

where $\lambda_{cov}$, $\lambda_{ent}$, and $\lambda_{sp}$ are the hyperparameters and $\mathbf{\Theta}_{gnn}$ denote overall parameters in $\hat{\phi}_f$ and $\hat{\phi}_e$, and the last term is $L_1$ loss to promote invariance to irrelevant contexts by preferring sparse weights.

**Inference**. The peer effect is obtained as $\hat{\delta}_i(\boldsymbol{\pi}_{-i}, \boldsymbol{\pi}'_{-i}) = \hat{f}(\pi_i, \boldsymbol{\pi}_{-i}, \mathcal{G}, \mathbf{X}, \mathbf{Z}) - \hat{f}(\pi_i, \boldsymbol{\pi}'_{-i}, \mathcal{G}, \mathbf{X}, \mathbf{Z}) = \hat{f}_y(\pi_i, \boldsymbol{\rho}_i, \mathbf{c}_i) - \hat{f}_y(\pi_i, \boldsymbol{\rho}'_i, \mathbf{c}_i)$, where $\hat{f}$ is the end-to-end EGONETGNN.

### 3.3 THEORETICAL ANALYSES OF EGONETGNN

**Expressiveness**. We perform a theoretical analysis of the expressive power of graph neural networks (GNNs) in capturing the causal network motifs proposed in the Yuan et al. (2021) paper. Building on previous research regarding the capacity of GNNs to count substructures (Chen et al., 2020), we demonstrate that existing message-passing GNN methods are not expressive enough to capture all causal network motifs. In contrast, our method is expressive to capture relevant causal network motifs. We defer the detailed theoretical framework and results, along with the relevant background, to Appendix A.4. We state our main result here.

**Proposition 2** (Expressiveness of EGONETGNN)**.** EGONETGNN is expressive enough to capture all dyad, open triad, closed triad, and open tetrad causal network motifs.

We sketch the proof by dividing the statement into following two claims. The details of the proofs of these claims are in Appendix A.4.

*Claim* 1. EGONETGNN is as expressive as standard MPGNN in capturing dyad, open triad, and open tetrad causal network motifs.

*Claim* 2. EGONETGNN also captures closed triad causal network motifs.

**Time complexity**. Our analysis of runtime complexity included in Appendix A.4.3 shows our method is, roughly on average, $\rho_{\mathcal{E}} \times avg(\mathbf{d})$ times more computationally expensive than standard MPGNNs, where $\rho_{\mathcal{E}}$ is the average edge density and $avg(\mathbf{d})$ is the average degree.

**Misspecification errors**. We extend Shalit et al. (2017)'s analyses of theoretical counterfactual prediction error bounds for the CFR model to study misspecification errors in the end-to-end EGONETGNN using the sequential error decomposition trick in Appendix A.4.4. By focusing on learning the expressive exposure mapping function, we are reducing its misspecification error directly.

## 4 EXPERIMENTS AND RESULTS

### 4.1 EXPERIMENTAL SETUP

**Dataset**. Similar to other works in causal inference, we rely on synthetic and semi-synthetic data. We consider three synthetic network models with a fixed number of nodes ($N = 3000$) with different data generating parameters and edge densities: (1) the Watts Strogatz (WS) network (Watts and Strogatz, 1998), which models small-world phenomena, (2) the Barabási Albert (BA) network (Albert and Barabási, 2002), which models preferential attachment phenomena, and (3) the Stochastic Block Model (SBM) that models community structures. We control the density of edges for BA and WS networks and the number of communities in the SBM network. We also use two real-world social networks, BlogCatalog and Flickr, with more realistic topology and attributes to generate treatments and outcomes. We defer additional details on data generation to Appendix A.5.

**Evaluation metrics**. To evaluate the performance of heterogeneous peer effect (HPE) estimation, we use the *Precision in the Estimation of Heterogeneous Effects* ($\epsilon_{PEHE}$) (Hill, 2011) metric defined as $\epsilon_{PEHE} = \sqrt{\frac{1}{n} \sum_i (\delta_i(\boldsymbol{\pi}_{-i}, \boldsymbol{\pi}'_{-i}) - \hat{\delta}_i(\boldsymbol{\pi}_{-i}, \boldsymbol{\pi}'_{-i}))^2}$, where $\delta_i(\boldsymbol{\pi}_{-i}, \boldsymbol{\pi}'_{-i})$ is true HPE and $\hat{\delta}_i(\boldsymbol{\pi}_{-i}, \boldsymbol{\pi}'_{-i})$ is the estimated HPE, where $\boldsymbol{\pi}'_{-i}$ denotes a counterfactual scenario where treatments of peers are flipped. $\epsilon_{PEHE}$ (lower better) measures the deviation of estimated HPEs from true HPEs. For each experimental result, we report the mean and standard deviation of $\epsilon_{PEHE}$ for 5 different simulations, i.e., data generation with different seeds. Evaluation across multiple simulations aims to demonstrate robustness across various possible patterns of peer treatment assignments or exposure conditions. For most experiments on semi-synthetic data, we use 3 random model initializations for each simulation.

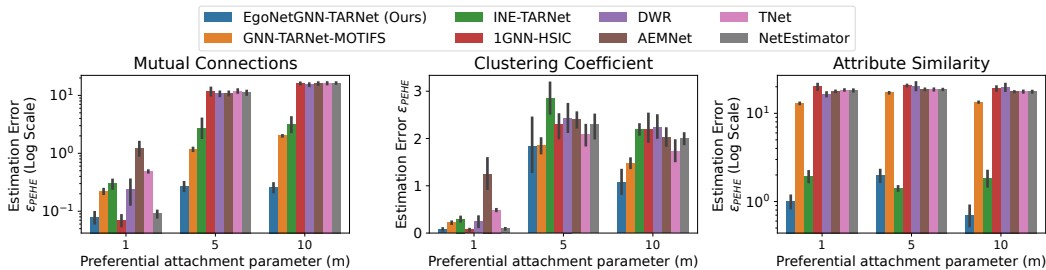

Figure 3: Peer effect estimation error for **Barabasi Albert** network when true peer exposure depends on mutual connections, clustering coefficient, and attribute similarity. Our method shows robust performance across different underlying peer influence mechanisms and edge densities (m).

Table 1: Mean and standard deviation (across 5 data simulations with 3 random model initializations each) of peer effect estimation error ($\epsilon_{PEHE}$) for different methods in **BlogCatalog** dataset for four settings when true peer exposure mechanisms depend on clustering coefficients, connected components, mutual connections, and attribute similarity.

| Mechanisms | Ours-TARNet | Ours-CFR$^+$ | GNN-Motifs | INE-TARNet | 1GNN-HSIC | DWR | AEMNet | TNet | NetEst | CauGramer |
|---|---|---|---|---|---|---|---|---|---|---|
| Clus. Coef. | $\underline{2.13}_{\pm1.9}$ | $\mathbf{0.95}_{\pm0.5}$ | $2.39_{\pm1.2}$ | $2.35_{\pm0.7}$ | $6.21_{\pm3.7}$ | $7.49_{\pm4.6}$ | $7.53_{\pm6.0}$ | $9.52_{\pm10.5}$ | $4.53_{\pm1.5}$ | $6.16_{\pm2.1}$ |
| Con. Comp. | $\mathbf{1.47}_{\pm0.9}$ | $\underline{1.50}_{\pm0.7}$ | $4.98_{\pm1.6}$ | $4.78_{\pm1.1}$ | $6.78_{\pm1.9}$ | $7.68_{\pm1.6}$ | $11.27_{\pm9.0}$ | $9.98_{\pm8.3}$ | $8.56_{\pm0.7}$ | $7.07_{\pm1.2}$ |
| Mut. Con. | $2.86_{\pm1.3}$ | $\mathbf{2.24}_{\pm1.6}$ | $2.81_{\pm1.3}$ | $\underline{2.50}_{\pm0.9}$ | $10.30_{\pm6.0}$ | $8.72_{\pm2.8}$ | $13.33_{\pm9.0}$ | $11.17_{\pm8.5}$ | $5.34_{\pm1.3}$ | $5.18_{\pm2.0}$ |
| Attr. Sim. | $3.95_{\pm2.7}$ | $\underline{3.65}_{\pm2.4}$ | $4.64_{\pm2.1}$ | $\mathbf{3.59}_{\pm1.8}$ | $15.25_{\pm4.7}$ | $17.96_{\pm3.7}$ | $14.10_{\pm5.0}$ | $14.60_{\pm5.0}$ | $11.71_{\pm2.2}$ | $14.45_{\pm5.7}$ |

Table 2: Mean and standard deviation (across 5 data simulations with one model initialization) HPE estimation error ($\epsilon_{PEHE}$ metric) for three variants of our method (original, without mask, and without feature encoder and mask) in the BlogCatalog (BC), Barabasi Albert (BA), and Watts Strogatz (WS) datasets for three true peer exposure mechanisms.

| Mechanism Network Model Variants | Mutual Connections | | | Clustering Coefficient | | | Attribute Similarity | | |
|---|---|---|---|---|---|---|---|---|---|
| | BC | BA | WS | BC | BA | WS | BC | BA | WS |
| Ours-TARNet | $2.61_{\pm1.0}$ | $\mathbf{0.20}_{\pm0.1}$ | $\mathbf{0.30}_{\pm0.1}$ | $\mathbf{1.71}_{\pm1.2}$ | $0.99_{\pm0.9}$ | $\mathbf{1.18}_{\pm0.8}$ | $4.79_{\pm3.2}$ | $\mathbf{1.23}_{\pm0.7}$ | $\mathbf{1.09}_{\pm1.1}$ |
| Ours (w/o mask) | $2.97_{\pm1.8}$ | $0.21_{\pm0.1}$ | $0.35_{\pm0.2}$ | $2.54_{\pm1.8}$ | $1.01_{\pm0.8}$ | $1.20_{\pm0.4}$ | $5.18_{\pm3.1}$ | $2.37_{\pm2.2}$ | $1.29_{\pm1.8}$ |
| Ours (w/o feat&mask) | $\mathbf{2.07}_{\pm1.3}$ | $0.27_{\pm0.2}$ | $0.31_{\pm0.1}$ | $2.11_{\pm0.8}$ | $\mathbf{0.97}_{\pm0.7}$ | $1.91_{\pm1.3}$ | $\mathbf{3.18}_{\pm1.9}$ | $13.73_{\pm2.8}$ | $13.85_{\pm4.0}$ |

Table 3: Evaluation of exposure representation, in terms of absolute correlation, in **BlogCatalog** data with no effect modification. The results for the learned peer exposure representation by our method is better (higher is better). We use the fraction of treated friends $z_i$ as baseline and the dimension of $\hat{\rho}, \hat{\rho}' \in [0, 1]^{d=2}$ with highest correlation is shown.

| Corr. | Clus. Coef. | Con. Comp. | Mut. Con. | Attr. Sim. | Corr. | Clus. Coef. | Con. Comp. | Mut. Con. | Attr. Sim. |
|---|---|---|---|---|---|---|---|---|---|
| $r(\hat{\rho}, \rho)$ | $\mathbf{0.81}_{\pm0.1}$ | $\mathbf{0.34}_{\pm0.3}$ | $\mathbf{0.73}_{\pm0.2}$ | $0.29_{\pm0.2}$ | $r(\hat{\rho}', \rho')$ | $\mathbf{0.85}_{\pm0.02}$ | $0.30_{\pm0.2}$ | $\mathbf{0.74}_{\pm0.1}$ | $0.50_{\pm0.1}$ |
| $r(z_i, \rho)$ | $0.17_{\pm0.1}$ | $0.12_{\pm0.1}$ | $0.09_{\pm0.03}$ | $0.28_{\pm0.2}$ | $r(z_i', \rho')$ | $0.41_{\pm0.2}$ | $0.14_{\pm0.1}$ | $0.09_{\pm0.1}$ | $\mathbf{0.61}_{\pm0.1}$ |

**Baselines**. We compare EGONETGNN with state-of-the-art (SOTA) peer estimation methods. NetEst (Jiang and Sun, 2022) and TNet (Chen et al., 2024) use the fraction of treated peers as peer exposure, but the estimator is based on adversarial learning and the doubly robust method, respectively, for robustness. DWR (Zhao et al., 2024) learns attention weights based on attribute similarity, and 1GNN-HSIC (Ma and Tresp, 2021) uses GNNs to summarize peer treatments as heterogeneous contexts while using homogeneous exposure. We also use the recently proposed GNN- and autoencoder-based automated exposure mapping approach (AEMNet) (Mao et al., 2025) and GNN- and transformer-based CauGramer (Wu et al., 2025) as baselines for estimating peer effects in our setup. We also consider INE-TARNet (Adhikari and Zheleva, 2025) adapted for peer effect estimation as a baseline, although it was developed for direct effect estimation. We include the GNN-TARNet-Motifs approach that considers manually extracted causal network motifs (Yuan et al., 2021) as peer exposure and TARNet as estimator (Shalit et al., 2017) as a strong baseline. We discuss hyperparameter tuning and model selection in Appendix A.6.

## 4.2 RESULTS

Next, we present results for experimental setups designed to answer five research questions (RQs).
**RQ1. How well do methods for peer effect estimation perform when peer exposure mechanisms depend on local neighborhood conditions?** In this setup, we evaluate the performance of peer effect estimators when the underlying peer exposure mechanism is unknown. We generate treatments and outcomes such that there is confounding due to a subset of node attributes and mean peer attributes. For the outcome generation, we consider five mechanisms for true peer exposure conditions where peer exposure is given by 1) the clustering coefficient between the treated peers, 2) the number of connected components among treated peers, and weighted fraction of treated peers with weights as 3) the square root of number of mutual connections, 4) attribute similarity, and 5) tie strength. Here, the unit's treatment acts as an effect modifier, where the peer exposure is doubled if the unit is treated. Figure 3 shows peer effect estimation error (y-axis), across five data simulations with fixed model initialization, when true peer exposure mechanisms depend on mutual connections, clustering coefficient, and attribute similarity in Barabasi Albert networks with three network generation parameters (x-axis), resulting in different edge densities (low to high). The preferential attachment parameter $m = 1$ produces a sparse star-topology network, lacking cycles or triangular structures. In this setting, all methods perform relatively well when peer exposure mechanisms depend on local structure because MPGNNs are expressive enough to capture star-shaped motifs. However, with increased edge density and more complex network topology, unlike our method, the baselines are not sufficiently expressive to capture underlying mechanisms and suffer significantly. The GNN-TARNet-Motifs (GTM) approach is expressive in capturing clustering coefficients, and both GTM and INE-TARNet approximate mutual connections. This is reflected in the performance, where GTM is competitive for the clustering coefficient peer exposure mechanism. EGONETGNN-TARNet outperforms the baselines except for INE-TARNet, which is competitive in a setting with the peer exposure mechanism dependent on attribute similarity. Figure 3 and other results in Appendix A.7 show that for unknown peer exposure mechanisms, our method is as expressive as or superior to the strongest baseline with significantly better performance for denser networks.

**RQ2. How reliable are the models for heterogeneous peer effect estimation in more realistic scenario?** RQ2 investigates the performance of the models using more realistic semi-synthetic networks and node attributes. In addition to confounding and heterogeneous peer influence, there is a more complex peer effect modification depending on whether the unit is treated and the values of the unit's attributes. Table 1 shows the mean and standard deviation of peer effect estimation error ($\epsilon_{PEHE}$), across five data simulations with three model initializations each, for different methods in the BlogCatalog (BC) dataset for four settings when true peer exposure mechanisms depend on clustering coefficients, connected components, mutual connections, and attribute similarity. The results show the robustness of EGONETGNN in a more realistic setting, where the variants of EGONETGNN are mostly the best performing ones. The baseline INE-TARNet is the most competitive, exhibiting slightly better performance than ours for the attribute similarity mechanism. However, like other methods, it still struggles when the underlying mechanisms involve complex local structures. In this setup, peer effects are heterogeneous due to the interaction of peer exposure conditions and effect modifiers, and our method is able to approximate them better than the baselines. Appendix A.8 presents additional experiments for this setup, including results for the Flickr dataset (Table 4), which is more challenging for the baselines. Table 7 in the Appendix shows that the variance in the results is primarily due to differences in data simulations rather than model initializations, as peer exposures resulting from some patterns of neighborhood treatment assignments can be easily captured by the models, while others cannot.

**RQ3. How do the components of EGONETGNN contribute to its robustness in estimating peer effects?** We conduct ablation studies to assess the contributions of masked weights and the feature encoder MLP. Table 2 displays the performance of three variants of EGONETGNN-TARNet (original, without the masked weights, and without the feature encoder and masked weights) across BlogCatalog (BC), Barabási Albert (BA), and Watts-Strogatz (WS) datasets. The results show that excluding masked weights can bias peer effect estimates due to the model's sensitivity to irrelevant contexts. Removing the feature encoder MLP limits EgoNetGNN's ability to capture mechanisms based on attribute similarity. Interestingly, for the semi-synthetic network, removing features produced even better results, most likely due to homophily, which results in attribute similarity that is almost homogeneous. As expected, for the peer exposure mechanisms relying on local structures, the model performs better when irrelevant features are ignored. Overall, these findings demonstrate

that the feature encoder MLP enhances expressiveness, while masked weights promote invariance to irrelevant contexts. Table 8 in the Appendix shows that the autoencoder component in our CFR$^+$ module preserves expressiveness and promotes robustness by comparing its performance with that of the original CFR, which does not include an autoencoder. Additionally, we analyze the EGONETGNN's sensitivity to the choices of peer exposure embedding dimension and noisy networks in Appendix A.9 and sensitivity to balance loss coefficient in Appendix A.10 (Table 9).

To mitigate the issue where peer exposure embedding ($\hat{\phi}_e$) captures a correlated pattern rather than the underlying mechanism, we perform model selection based on prediction loss and coverage loss in a 20% validation dataset. The idea is that choosing a correlated pattern rather than a true one is akin to overfitting. In Table 10, we evaluate this model selection strategy against the one based on prediction loss only. The results show model selection utilizing coverage loss is more robust, which could be because the coverage loss aims to prevent the equivalent of *mode collapse*, where the distribution of output peer exposure representation is limited.

**RQ4. How well are the underlying mechanisms captured by the learned exposure mapping function?** In Table 3, we directly compare the (absolute) Pearson correlation coefficient $r$ (higher is better) between the learned peer exposure representation, $\hat{\rho}$ and $\hat{\rho}'$, and the actual peer exposure under four different mechanisms. Compared to the commonly used fraction of treated friends baseline, learned peer exposures are informative of true peer exposures for mechanisms involving local structure. For attribute similarity mechanism, the baseline approach maybe competitive due to homophily, where attributes of neighbors are homogeneous.

**RQ5. How well does EGONETGNN perform under homogeneous exposure and imperfect conditions when the model assumptions are violated?** First, we evaluate the models in the simplest setting, where all baselines make the correct exposure mapping function assumption, i.e., true peer exposure depends on the fraction of treated peers. Table 11 shows that variants of our model remain superior to the baselines even when they make correct assumptions about the underlying peer exposure mechanisms, as they struggle in settings such as complex effect modifications and arbitrary counterfactual spaces (i.e., flipped counterfactuals). We show the trade-off between computation time, memory requirement, and performance for this simple setting in Table 12 to show how our method gains robustness with extra but manageable computation time. Second, we evaluate the models in the setting with censored or noisy features by randomly zeroing out 10% of the features and adding Gaussian noise. Finally, we evaluate the models in the presence of confounding and interference from two-hop neighbors to examine how our model performs when its assumptions are violated. Tables 13 and 14 show that, although the magnitude of error is increased, our methods are competitive with or better than the prominent baselines in such imperfect settings.

## 5 DISCUSSION, LIMITATIONS & FUTURE WORK

Our work motivates the problem of learning exposure mapping function for peer effect estimation and proposes EGONETGNN for addressing unknown peer influence mechanisms involving local neighborhood conditions. Our theoretical analysis and experimental results demonstrate increased expressiveness of EGONETGNN to capture complex local neighborhood exposure conditions. We have designed EGONETGNN to promote invariance to irrelevant contexts, and output a low-dimensional peer exposure embedding with bounded and balanced representation to partially mitigate issue of potential violation of the positivity assumption with continuous treatment or exposure. The empirical results have shown the effectiveness of EGONETGNN in many peer effect estimation settings.

**Limitations & Future Work.** Our main theoretical results are on the expressiveness of the GNN to capture complex causal network motifs. We also discuss counterfactual prediction error bounds by decomposing to misspecification errors in features mapping, exposure mapping, and outcome models. Yet establishing theoretical bounds and asymptotic properties with complex GNNs for heterogeneous causal effect estimation is still a developing research area (Khatami et al., 2024) and important future direction, and it is not within the scope of our current work. This work can be extended to incorporate other network effects like direct effects and total effects. The increased expressiveness and robust peer effect estimates of our model come with the trade-off of a slightly longer runtime to process ego networks. Future work could consider relaxing the assumption of interference from immediate peers while addressing the scalability. Our work relies upon a reliable attributed network as input, but future research should consider capturing expressive representations in noisy networks. Appendix A.1 discusses societal impacts, scalability, and plausibility of assumptions.

## REPRODUCIBILITY STATEMENT

To support reproducibility, we release the complete codebase and experimental procedures in the repository `https://github.com/edgeslab/EgoNetGNN`. For all the experiments, we have repeated them at least *five* times, with different data generation and/or model initialization. We have provided the details of the data generation process (Sec. 4.1 and Appendix A.5). Also, the details of the configurations and setups for replicating our results are in Appendix A.6 and in the documentation of the parameters in the source code.

## ACKNOWLEDGMENT

This research was supported in part by the U.S. National Science Foundation under Grant No. 2047899 and by the Defense Advanced Research Projects Agency (DARPA) under Contract No. HR001121C0168. Any opinions, findings, and conclusions or recommendations expressed in this material are those of the author(s) and do not necessarily reflect the views of the NSF, DARPA, or the U.S. Government.

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

# A    APPENDIX

## A.1    DISCUSSION

**Societal impacts**. The implications of our work include identifying unit-level peer effects and discovering subpopulations with heterogeneous peer effects. The potential societal impacts could include the development of targeted interventions or the identification of policies that enhance desired outcomes in social networks.

**Plausibility of neighborhood interference assumption**. Neighborhood interference (Assumption 2 in Sec. 2) is a common simplifying assumption and can be realistic in situations where peer interference is mediated by immediate neighbors or diminishes quickly for non-immediate neighbors. However, there could be some situations where interference could occur between peers beyond immediate neighbors. If we assume such interference is mediated via immediate neighbors, then we could model it by stacking multiple exposure mapping function learning layers, where the subsequent layers would summarize the exposures of neighbors. Another alternative is to use the K-hop ego network with edge existence and/or hop distance as additional node features. The former approach may be more scalable than the latter one because the K-hop neighborhood can grow rapidly. Ideas from recent works to infer unknown interference structure (Wu et al., 2025; Lin et al., 2024) could be adopted in conjunction with our approach of learning expressive peer exposure representations. Our method can be extended to integrate attention or transformer-based mechanisms in either feature mapping learning or exposure mapping learning by replacing/modifying MPGNN architectures. While we assume a reliable network structure is provided as input, our experiments with noisy networks reveal that EGONETGNN performs reliably well with imperfect data.

**Plausibility of unconfoundedness assumption**. Following existing work in the intersection of causal reasoning and representation learning (Shalit et al., 2017; Shi et al., 2019; Ma et al., 2022; Wu et al., 2025), we assume causal identification conditions are met and focus on expressive representation learning to mitigate model misspecification errors. Unconfoundness is a strong and untestable assumption and requires sufficiency of observed network contexts and expressiveness of their representation. While we assume the sufficiency of observed contexts, we make an effort to satisfy the expressiveness of representation by considering all network contexts, like node attributes, edge attributes, and network structure. If the presence of unobserved confounding cannot be ruled out, alternative causal identification approaches like proximal causal inference (Tchetgen et al., 2020) or double negative controls (Miao et al., 2024), front-door criteria (Pearl, 2009), and instrumental variables (Angrist et al., 1996) should be considered. Although a randomized experiment can remove unobserved confounding between unit treatments and the outcome, peer exposure conditions may not be randomized directly, and confounding could exist even for experiments unless the unconfoundedness assumption is made and observed network contexts are controlled for. So, an interesting future direction could be to explore alternative identification conditions.

**Scalability**. Although EGONETGNN is more expressive, it has additional computational costs. A few ways to address large runtime and/or memory usage could be sampling ego networks to reduce the training set or sampling the neighborhood within a K-hop ego network. In Appendix A.8 (Table 6), our experiments with a randomly augmented network show that the performance does not degrade significantly for our method with the removal of edges. From an implementation point of view, we can parallelize our framework easily to exploit the power of GPUs. More specifically, there are two components in our framework. The feature mapping GNN takes the entire network at once to learn an embedding with an L-layer GNN. Subsequently, EgoNetGNN batches B nodes with their neighbor nodes and a mapping of which edges belong to which node in the batch. This batching can be parallelized to improve the overall efficiency.

**Representation balancing in CFR**. We note that explicitly enforcing IPM balancing (using approaches such as the Wasserstein distance) for the ideal condition, $\mathbb{P}(\mathbf{c}|\mathbf{p}, t) \approx \mathbb{P}(\mathbf{c}|\mathbf{p}', t')$, is non-trivial at best and computationally infeasible at worst. The challenges arise primarily because the peer exposures ($\mathbf{p}$ versus $\mathbf{p}'$) are multi-dimensional continuous values. There may be several exposure conditions with mostly limited samples, which complicates the calculation of the Wasserstein distance (optimal transport) and may render it difficult or unreliable, even when using modern methods. Our balancing technique is a computation-friendly and useful heuristic approach (as evidenced by the experiments), where for each stratum of peer exposure condition, we want balanced covariates across different treatment groups. It does not perform the ideal balancing scenario. However, even in

network experiments where treatments are randomized, such balancing is not possible because peer exposure conditions depend not only on peer treatments but also on other covariates; therefore, the randomization for peer exposures is not achieved. We instead control for different possible network contexts (derived from the observational attributed network), similar to regression adjustment, to deal with the imbalance of peer exposure conditions.

**Theoretical guarantees and confidence intervals with complex GNNs**. While complex GNN and transformer architectures are very powerful in capturing confounding variables, effect modifications, and unknown influence mechanisms in complex network data, they lack interpretability and theoretical guarantees of consistency or convergence. As discussed in our paper, theoretical properties of complex GNNs have been shown in simpler settings where there is a homogeneous exposure mapping function and for average treatment effects (Khatami et al., 2024; Chen et al., 2024). Establishing theoretical guarantees in the setting with unknown exposure mapping functions and heterogeneous effects is an important future direction, but it is outside the scope of our paper. Instead, our main theoretical results in this paper are on the expressiveness of the GNN to capture complex underlying influence mechanisms. In this work, we focus on point estimates without confidence intervals. For real-world data, techniques like bootstrapping and random model initializations could be used to obtain a measure of uncertainty similar to confidence intervals. Like ours, other works have used point estimates to present empirical evidence to support robustness in complex interference settings, such as using GNN on hypergraphs to model group interactions (Ma et al., KDD 2022) and using Graph Transformers to model unknown interference structure (e.g., CauGramer (Wu et al., ICLR 2025)), leaving theoretical results on error bounds with complex models to future work. We hope our work could spur future research directions and collaborations to address these limitations. Our framework is flexible enough to be adapted to utilize unbiased estimators like the Horvitz-Thompson (HT) estimator or the Targeted Maximum Likelihood Estimation (TMLE) estimator for causal estimation, making it still appealing for practitioners.

## A.2 RELATED WORK

Research in causal inference under interference has focused on estimating three main causal effects of interest, referred to as network effects: direct effects induced by a unit's own treatment, peer effects induced by treatment of other units, and total effects induced by both the unit's and others' treatment (Hudgens and Halloran, 2008). These network effects are estimated as average effects (e.g., (Arbour et al., 2016; Ugander et al., 2013)) for the entire population or as heterogeneous effects (e.g., (Forastiere et al., 2021; Bargagli-Stoffi et al., 2025)) for specific subpopulations or contexts. Our work focuses on heterogeneous peer effect estimation. Most methods for estimating heterogeneous or individual-level causal effects under interference, including peer effects, assume peer exposure is binary (Bargagli-Stoffi et al., 2025) or homogeneous, e.g., based on fraction of treated peers (Jiang and Sun, 2022; Ogburn et al., 2022; Cai et al., 2023; Chen et al., 2024). These methods assume a homogeneous or known exposure mapping function and focus on enhancing network effect estimation by adapting techniques like adversarial training (Jiang and Sun, 2022), propensity score reweighting (Cai et al., 2023), double machine learning (Khatami et al., 2024), doubly robust estimation (Leung and Loupos, 2022), targeted maximum likelihood estimate (Ogburn et al., 2022), and targeted learning (Chen et al., 2024).

Recent research has looked into more complex functions of peer exposure, allowing for heterogeneous peer influence, in which different peers can have varying degrees of influence. Some of these works refer to heterogeneous peer influence as heterogeneous interference (Qu et al., 2021; Zhao et al., 2024; Lin et al., 2023). Forastiere et al. (2021) considered peer exposure as a weighted fraction of treated peers using known edge attributes as weights. Lin et al. (2023) consider heterogeneity due to multiple entities types and Qu et al. (2021) considered heterogeneity due to known node attributes for defining peer exposure. Tran and Zheleva (2022) studied peer effect estimation with linear threshold peer exposure model but different unit-level threshold could be vary for different units capturing heterogeneous susceptibilities to the influence. Zhao et al. (2024) used attention weights derived based on the similarities of the units' covariates to determine peer exposure as the weighted sum of treated peers. Yuan et al. (2021) capture peer exposure with features based on counts of different causal network motifs, i.e., recurrent subgraphs in a unit's ego network with treatment assignments as attributes. Ma and Tresp (2021) consider homogeneous peer exposure based on fraction of treated peers but they summarize the covariates of treated peers using a graph neural network (GNN) to capture heterogeneous contexts involving treatment assignments. Unlike our work, none of these

studies has explicitly studied the issue of automatically learning the exposure mapping functions to define peer exposure representation while capturing the underlying influence mechanisms.

Ma and Tresp (2021) learn heterogeneous contexts based on peer treatments but not the exposure mapping function or the peer exposure representation. Zhao et al. (2024) obtain single-dimension peer exposure embedding using a weighted sum of treated peers with attention weights derived from the cosine similarity of feature embeddings. Although Zhao et al. (2024) use attention weights to define peer exposure, they assume a specific exposure mapping function, and it cannot adapt according to the underlying peer influence mechanism. Adhikari and Zheleva (2025) use GNNs to learn peer exposure embedding by addressing unknown peer influence mechanisms, but their scope is limited to direct effect estimation, i.e., the effect of a unit's own treatment. Specifically, Adhikari and Zheleva (2025) learn a multi-dimensional peer exposure embedding using a weighted fraction of treated peers with feature embeddings and a second-order adjacency matrix as weights. Ma et al. (2022) employ similar method like Ma and Tresp (2021) for hypergraphs to model heterogeneity due to model group interactions. The idea is to learn a summary function and representation equivalent to the exposure mapping function and peer exposure using a hypergraph convolution network and attention mechanism. However, they assume the learned representation is expressive enough to capture the underlying influence mechanism. In this work, we do not make such an assumption and evaluate how well the learned peer exposure representation captures the underlying influence mechanisms.

Neural networks (NNs) (Shalit et al., 2017; Im et al., 2021; Shi et al., 2019) and, recently, graph neural networks (GNNs) (Jiang and Sun, 2022; Cai et al., 2023; Chen et al., 2024; Khatami et al., 2024) have been widely utilized for end-to-end learning of *feature mapping function* and *counterfactual outcome model* or *effect estimator*. A feature mapping function maps raw features to feature embedding to capture potential confounders and effect modifiers. A counterfactual outcome model (Shalit et al., 2017; Ma and Tresp, 2021) predicts counterfactual outcomes for different levels of treatment, while an effect estimator (Shi et al., 2019; Chen et al., 2024) directly learns the causal effect of interest. Only a few studies have considered learning the exposure mapping function (Mao et al., 2025) or peer exposure embedding (Adhikari and Zheleva, 2025; Zhao et al., 2024). Lin et al. (2024) consider a setting with an unknown network and interference structure and propose an approach to first infer network structure and represent peer exposure for direct effect estimation. Unlike their work, our settings focus on peer effect estimation with observed network structure but unknown peer exposure mechanisms that manifest due to local neighborhood contexts.

Sävje (2024) advocates for interpretable but possibly misspecified exposure mappings and characterizes causal estimation errors due to misspecified exposure mappings, but follow-up research (Auerbach et al., 2024) has highlighted the importance of capturing underlying interference mechanisms in policymaking. More recently, Mao et al. (2025) have explored the use of GNNs with autoencoders and clustering to learn discrete exposure conditions and their probabilities, aiming to estimate overall causal effects in networks. Similarly, Wu et al. (2025) utilize GNNs with Transformers to model unknown interference from K-hop neighborhood. Their identifiability assumption relies on capturing unit and peer covariates, while our identifiability assumption relies on capturing all attributed network contexts, including structure and edge attributes. These works use off-the-shelf message passing GNNs (like GCN and GIN) and lack expressiveness to capture mechanisms involving local neighborhood structure. Prior research (Xu et al., 2018; Chen et al., 2020) on the expressiveness of GNNs has shown that popular GNN architectures lack expressiveness to count subgraphs. On the other hand, counts of subgraphs like causal network motifs are rich features that could capture underlying influence mechanisms due to local neighborhood structure (Yuan et al., 2021). Counting such subgraphs can be computationally expensive, and they may not be able to capture every local structure. We design EGONETGNN to excel in counting attributed triangle subgraphs, enhancing its expressiveness to capture underlying mechanisms involving neighborhood contexts.

### A.3 CAUSAL INFERENCE ASSUMPTIONS AND IDENTIFICATION OF PEER EFFECTS

A fundamental prerequisite for causal identification is the consistency assumption, which enables equivalence among counterfactual, interventional, and factual outcomes.

**Assumption 4** (Consistency under interference). The underlying outcome generation is independent of the treatment assignment mechanisms (i.e., hypothetical, experimental, or natural). For a unit $v_i$, if $t_i = \pi_i$ and $\mathbf{t}_{-i} = \boldsymbol{\pi}_{-i}$, then $y_i(t_i = \pi_i, \mathbf{t}_{-i} = \boldsymbol{\pi}_{-i}) = y_i$.

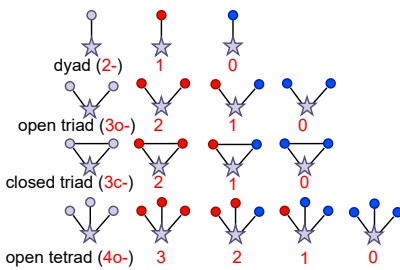

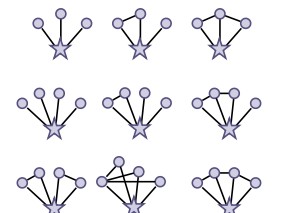

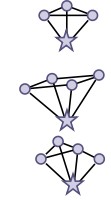

Examples of higher-order network motifs that **can** be captured by EgoNetGNN

Examples of higher-order network motifs that **cannot** be capturedby EgoNetGNN

Figure 4: Example causal network motifs considered by Yuan et al. (2021). Stars represent ego nodes and circles represent their peers. The red circles indicate treated nodes and blue circles indicate control nodes. The gray shapes indicate nodes that could either be treated or control. Here, the characters in red indicate a particular causal network motif (e.g., 3c-2 indicate closed triad with 2 treated peers).

Figure 5: Examples of higher-order network motifs with four and five nodes. Stars represent ego nodes and circles represent their peers. The gray shapes indicate nodes with any treatment assignment. If the subgraph of a network motif, after removing edges connected to the ego node, forms a tree, then our model is expressive enough to capture the network motif and the corresponding causal network motifs. A network motif is a subgraph without any attributes, whereas a causal network motif is a subgraph that includes peer treatment assignments as attributes.

Positivity is another standard assumption in causal inference that requires every unit $v_i$ to have non-zero probability of being assigned every possible unit treatment and peer exposure conditions.

**Assumption 5** (Positivity). There is a non-zero probability of unit treatment and peer exposure conditions for all possible contexts $\mathbf{c}_i$, i.e., $\mathbb{P}(\mathbf{t}_i, \mathbf{p}_i | \mathbf{c}_i) > 0$, for every level of $\mathbf{t}_i$ and $\mathbf{p}_i$, where $\mathbb{P}$ is the probability density function.

The proof of Proposition 1 is as follows.

*Proof.* Our causal estimand of interest (Eq. 2) is as follows:

$$\delta_i(\boldsymbol{\pi}_{-i}, \boldsymbol{\pi}'_{-i}) = \mathbb{E}[\mathbf{y}_i(\mathbf{t}_i = \pi_i, \mathbf{p}_i = \phi_e(\boldsymbol{\pi}_{-i}, \mathcal{G}, \mathbf{X}, \mathbf{Z})) | \mathbf{c}_i] - \mathbb{E}[\mathbf{y}_i(\mathbf{t}_i = \pi_i, \mathbf{p}_i = \phi_e(\boldsymbol{\pi}'_{-i}, \mathcal{G}, \mathbf{X}, \mathbf{Z})) | \mathbf{c}_i].$$

Due to unconfoundedness assumption (Assumption 3), unit treatment and peer exposure conditions are independent of counterfactual outcome conditioned on network contexts $\mathbf{c}_i$. This allows us to rewrite the estimand as:

$$\delta_i(\boldsymbol{\pi}_{-i}, \boldsymbol{\pi}'_{-i}) = \mathbb{E}[\mathbf{y}_i(\mathbf{t}_i = \pi_i, \mathbf{p}_i = \phi_e(\boldsymbol{\pi}_{-i}, \mathcal{G}, \mathbf{X}, \mathbf{Z})) | \mathbf{t}_i = \pi_i, \mathbf{p}_i = \phi_e(\boldsymbol{\pi}_{-i}, \mathcal{G}, \mathbf{X}, \mathbf{Z}), \mathbf{c}_i] -$$
$$\mathbb{E}[\mathbf{y}_i(\mathbf{t}_i = \pi_i, \mathbf{p}_i = \phi_e(\boldsymbol{\pi}'_{-i}, \mathcal{G}, \mathbf{X}, \mathbf{Z})) | \mathbf{t}_i = \pi_i, \mathbf{p}_i = \phi_e(\boldsymbol{\pi}_{-i}, \mathcal{G}, \mathbf{X}, \mathbf{Z}), \mathbf{c}_i].$$

Here, Assumption 1 ensures introducing new terms related to treatment and peer exposure in the conditional does not affect existing set of contexts because they are measured pre-treatment. Similarly, Assumption 2 makes the sufficiency of learned representation requirement in unconfoundedness assumption more plausible. Next, the consistency assumption allows replacing the counterfactual outcome with observed outcome, i.e.,

$$\delta_i(\boldsymbol{\pi}_{-i}, \boldsymbol{\pi}'_{-i}) = \mathbb{E}[\mathbf{y}_i | \mathbf{t}_i = \pi_i, \mathbf{p}_i = \phi_e(\boldsymbol{\pi}_{-i}, \mathcal{G}, \mathbf{X}, \mathbf{Z}), \mathbf{c}_i] - \mathbb{E}[\mathbf{y}_i | \mathbf{t}_i = \pi_i, \mathbf{p}_i = \phi_e(\boldsymbol{\pi}_{-i}, \mathcal{G}, \mathbf{X}, \mathbf{Z}), \mathbf{c}_i].$$

Assumption 1 also ensures consistency assumption is satisfied because the treatments are not mutable. This estimation above is tractable from observational or experimental data because of positivity assumption and the causal effects can be identified. □

### A.4 THEORETICAL ANALYSES OF EGONETGNN

#### A.4.1 PRELIMINARIES

**Causal network motifs**. Yuan et al. (2021) proposed causal network motifs as important features to capture peer exposure accounting for local neighborhood conditions. Causal network motifs are attributed subgraphs with peer treatments as attributes. Figure 4 shows four categories of causal

network motifs: dyads, open triads, closed triads, and open tetrads. In the figure, stars represent ego nodes and circles represent their peers. The red circles indicate treated nodes and blue circles indicate control nodes. The gray shapes indicate nodes that could either be treated or control.

**Message passing graph neural networks (MPGNNs).** The message-passing graph neural network (MPGNN) is a generic GNN model that incorporates several standard GNN architectures and relies on local aggregations of information within graphs (Chen et al., 2020). For a graph $G(V, E, \mathbf{X}, \mathbf{Z})$, an MPGNN with $L$ layers is defined iteratively with aggregate function $AGG^l$ and update function $U^l$ as follows:

$$h_i^l = U^l(h_i^{l-1}, AGG_{j \in \mathcal{N}_i}^l(\Theta^l(h_j^{l-1}, h_i^{l-1}, Z_{ij}))), \tag{11}$$

where $\mathcal{N}_i$ denotes neighbors of unit $v_i$ and $\Theta^l$ denote learnable parameters like multi-layer perceptron. To obtain the hidden state at the $l^{th}$ layer, a local aggregation of the previous layer's hidden states ($h_j^{l-1}$ and $h_i^{l-1}$) and, optionally, edge attributes $Z_{ij}$ is performed and then combined with $h_i^{l-1}$. The hidden states are initialized as node attributes, i.e., $h_i^0 = X_i$. Typically, in various GNN architectures, the update and aggregation functions are chosen as part of architecture design.

**Expressiveness of MPGNNs in counting substructures.** Here, we summarize the results obtained by Chen et al. (2020) that are relevant to our theoretical analysis. We list their findings after defining relevant concepts.

**Definition 2** (Subgraph). A *subgraph* $G^{[S]}(V^{[S]}, E^{[S]})$ of a graph $G(V, E)$ consists of subsets of its nodes, i.e., $V^{[S]} \subseteq V$ and edges, i.e., $E^{[S]} \subseteq E$.

**Definition 3** (Induced subgraph). A *induced subgraph* $G^{[S']}(V^{[S']}, E^{[S']})$ of a graph $G(V, E)$ consists of subset of its nodes, i.e., $V^{[S']} \subseteq V$ and all edges between nodes $V^{[S']}$, i.e., $E^{[S']} = E \cap V^{[S']}$.

All induced subgraphs are subgraphs but reverse is not true. For example, all causal network motfis are induced subgraphs (and subgraphs) of the original graph. An open triad motif is a subgraph, but not an induced subgraph, of a closed triad motif.

**Definition 4** (Star-shaped pattern). A pattern $G^{[P]}(V^{[P]}, E^{[P]})$ is a star-shaped pattern if it can be represented by a tree structure.

**Definition 5** (Connected pattern). A pattern $G^{[P]}(V^{[P]}, E^{[P]})$ is a connected pattern if it **cannot** be represented by a tree structure.

For example, a closed triad motif is a connected pattern and dyads, open triads, and open tetrads are star-shaped patterns.

Chen et al. (2020) obtain the following results on the expressiveness of MPGNNS for counting substructures.

**Corollary 3.4.** (Chen et al., 2020) MPGNNs cannot *induced-subgraph-count* any *connected pattern* with 3 or more nodes.

**Theorem 3.5.** (Chen et al., 2020) MPGNNs can perform *subgraph-count* of *star-shaped patterns*.

### A.4.2 EXPRESSIVENESS OF EGONETGNN

Here, we demonstrate that standard MPGNNs lack the expressiveness to capture closed triad motifs, and our model addresses this limitation.

Without loss of generality, assume node attributes for each node $v_i$ are $< 1, T_i >$ and constant edge attributes $< 1 >$.

**Definition 6** (Expressiveness in counting causal network motifs). Let $\mathcal{G}$ be a space of graphs. A representation by an MPGNN $f$ is expressive in counting causal network motif $G^{[P]}$ if, for all ego networks $G^{[1]}, G^{[2]} \in \mathcal{G}$, distinct counts, i.e., $C_I(G^{[1]}, G^{[P]}) \neq C_I(G^{[2]}, G^{[P]})$, get distinct representations, i.e., $f(G^{[1]}) \neq f(G^{[2]})$, where $C_I$ returns induced-subgraph-count of pattern $G^{[P]}$.

**Restating Proposition 2 (Expressiveness of EGONETGNN):** EGONETGNN is expressive enough to capture all dyad, open triad, closed triad, and open tetrad causal network motifs.

*Proof.* We proceed the proof by dividing the statement into following two claims.

**Restating Claim 1:** EGONETGNN is as expressive as standard MPGNN in capturing dyad, open triad, and open tetrad causal network motifs.

*Proof.* The dyad, open triad, and open tetrad causal network motifs are star-shaped patterns, and these patterns can be counted by standard MPGNNs (Chen et al. (2020)'s Theorem 3.5.). Our model employs MPGNN (refer Eq. 4 and Figure 2) on a transformed graph, where all edges connected to the ego node are removed, and the corresponding edge attributes from the removed edges are included as node attributes in the transformed graph. We need to show that this transformation preserves the expressiveness to capture dyad, open triad, and open tetrad causal network motifs. The dyad, open triad, and open tetrad causal network motifs are transformed into subgraphs with isolated one, two, and three nodes, respectively, in the transformed ego network. MPGNN in the transformed graph can perform a subgraph count of patterns with k isolated nodes because they are subgraphs of star-shaped patterns with an empty set of edges. Furthermore, the addition of new attributes does not affect the expressiveness because these attributes are added as additional feature dimensions. Hence, our model is as expressive as standard MPGNN for capturing dyad, open triad, and open tetrad causal network motifs. □

**Restating Claim 2:** EGONETGNN also captures closed triad causal network motifs.

*Proof.* The closed triad causal network motifs are connected patterns of three nodes and these patterns cannot be counted by standard MPGNNs (Chen et al. (2020)'s Corollary 3.4.). Due to the construction of the ego network, all the edges with the ego node are removed, and the closed triads are transformed to dyads in the transformed ego network. These dyads can be counted by node aggregation (refer Eq. 4), which is an MPGNN employed in the ego network. Therefore, EGONETGNN captures closed triad causal network motifs. □

□

**Higher-order causal network motifs and attributed causal network motifs.** Here, we show how our model is superior to the approach of counting predetermined causal network motifs by discussing EGONETGNN's ability to capture relevant causal network motifs including higher-order and attributed causal network motifs. Proposition 2 showed our model is as expressive as the approach of counting predetermined causal network motifs considered by Yuan et al. (2021). In general, if the subgraph of a network motif, after removing edges connected to the ego node, forms a tree, then EGONETGNN is expressive enough to capture the network motif and the corresponding causal network motifs. Figure 5 depicts some examples of higher-order motifs with four and five nodes. EGONETGNN, with depths of $L = 2$ and $L = 3$ (refer Eq. 4), is expressive enough to capture most higher-order motifs with four and five nodes, respectively. Only if the network motifs consist of a cycle without the involvement of the ego node, then EGONETGNN is not expressive enough to capture it. Furthermore, compared to predetermined causal network motifs, EGONETGNN can accommodate motifs with additional node and edge attributes. Incorporating node and edge attributes will not reduce the expressiveness of counting original causal network motifs because these attributes are added as additional feature dimensions.

### A.4.3 TIME COMPLEXITY OF EGONETGNN-TARNET

Typically, the complexity of a standard MPGNN (e.g. GCN), is $O(NLF^2 + L|E|F)$, where $N$, $|E|$, $L$, and $F$ are the number of nodes, edges, GNN layers, and the dimensionality of feature embeddings, respectively (Blakely et al., [n. d.]). In our model, the feature mapping MPGNN (refer to Eq. 3) has the time complexity of $O(d_{\Theta}N{F_x}^2)$ for ego feature embedding module $\Theta_0(\mathbf{X_i})$, where $d_{\Theta}$ is the depth of MLP and $F_x$ is the dimensionality of node feature embedding, and $O(Ld_{\Theta}|E|F^2 + L|E|F)$ for peer feature embedding and aggregation, where $F = F_x + F_z$ is the dimensionality of node and edge feature embeddings. For node aggregation (refer to Eq. 4), we extract ego network for each node and perform neighborhood aggregation. Therefore, the time complexity is $O(NL|\bar{E}_{max}|F)$, where $|\bar{E}_{max}|$ is the number of maximum edges in the ego network. For subsequent masking and exposure encoding MLP, the time complexity is $O(Nd_{MLP}|\bar{E}_{max}|F^2)$, where $d_{MLP}$ is the depth considering overall MLPs.

Assuming a single-layer MPGNN with $F << N < |E|$, for simplicity, a standard MPGNN scales linearly with the number of edges, i.e., $O(|E|)$ or $O(N \times avg(D))$, where $avg(D)$ is the average degree. Similarly, for EGONETGNN the time complexity simplifies to $O(N \times |\bar{E}_{max}|)$. In the worst case, $|\bar{E}_{max}| = max(D)^2$, where $max(D)$ is the maximum degree in the network $G(V, E)$. However, since networks are generally sparse, the approximate runtime complexity for networks with uniform degree (e.g., Watts Strogatz network or Stochastic Block Model network) is $O(N \times P_e \times avg(D)^2)$, where $P_e$ is density of edges. So, our method is approximately $P_e \times avg(D)$ times more computationally expensive than standard MPGNNs. On the other hand, the time complexity for counting predetermined causal network motifs with $K$ nodes is $O(Nmax(D)^{K-1})$, assuming access to O(1) adjacency set and adjacency matrix. This approach scales poorly with higher-order motifs and EGONETGNN mitigates the problem by capturing most higher-order motifs with the same computational cost.

### A.4.4 COUNTERFACTUAL OUTCOME PREDICTION ERROR BOUNDS FOR EGONETGNN

Our work utilizes Shalit et al. (2017)'s TARNet and CFR estimators, adapted to network settings, for estimating heterogeneous peer effects in both observational and experimental data. Their analysis shows the $PEHE$ metric is bounded by factual ($F$), i.e., supervised learning and counterfactual ($CF$) prediction error, i.e., $\epsilon_{PEHE}(\hat{f}_y) \leq 2(\epsilon_{CF}(\hat{f}_y) + \epsilon_F(\hat{f}_y) - 2\sigma_y^2)$, where $\sigma_y^2$ is the variance of the outcome. These prediction errors or biases incorporate Sävje (2024)'s definition of exposure mapping specification errors along with feature representation errors and outcome prediction errors.

Moreover, Shalit et al. (2017) show that the bound for counterfactual prediction error (which cannot be measured in the real world) depends on the Integral Probability Metric (IPM) measure of distance between treatment and control group distribution, which implies $\epsilon_{PEHE}(\hat{f}_y) \leq 2(\epsilon_F^{t_i=1}(\hat{f}_y) + \epsilon_F^{t_i=0}(\hat{f}_y) + \alpha IPM(\{h_i^{emb} : t_i = 1\}, \{h_i^{emb} : t_i = 0\}) - 2\sigma_Y^2)$, where $t_i = \pi_i$ denotes conditioning, $h_i^{emb} = \Theta_{emb}(\hat{c}_i || \hat{\rho}_i)$, and $||$ denotes concatenation. To study how misspecification errors of EgoNetGNN propagate to the factual prediction error, we can substitute the oracle values and estimated values (denoted with hat) and further decompose the errors by using sequential error decomposition trick, i.e.,

$$\epsilon_F^{t_i=\pi_i}(\hat{f}_y) = \mathbb{E}[(\hat{y}_i - y_i)^2]$$

$$\hat{y}_i - y_i = \hat{f}_y(\pi_i, \hat{\rho}_i, \hat{c}_i) - f_y(\pi_i, \rho_i, c_i)$$

$\hat{y}_i - y_i = \epsilon_y + \epsilon_e + \epsilon_f$, where $\epsilon_y$ captures error due to learned outcome prediction module using learned representations, i.e.,

$$\epsilon_y := \hat{f}_y(\pi_i, \hat{\rho}_i, \hat{c}_i) - f_y(\pi_i, \hat{\rho}_i, \hat{c}_i),$$

$\epsilon_e$ captures error due to exposure mapping misspecification using learned feature representation but true outcome prediction module, i.e.,

$$\epsilon_e := f_y(\pi_i, \hat{\rho}_i, \hat{c}_i) - f_y(\pi_i, \rho_i, \hat{c}_i),$$

and, finally, $\epsilon_f$ captures error due to feature mapping misspecification but true exposure and outcome prediction function, i.e.,

$$\epsilon_f := f_y(\pi_i, \rho_i, \hat{c}_i) - f_y(\pi_i, \rho_i, c_i).$$

By plugging these decomposed errors in the factual prediction loss, we get,

$$\epsilon_F^{t_i=\pi_i}(\hat{f}_y) = \mathbb{E}[(\epsilon_y + \epsilon_e + \epsilon_f)^2]$$

$$= \mathbb{E}[\epsilon_y^2] + \mathbb{E}[\epsilon_e^2] + \mathbb{E}[\epsilon_f^2] + 2(\mathbb{E}[\epsilon_y\epsilon_e] + \mathbb{E}[\epsilon_e\epsilon_f] + \mathbb{E}[\epsilon_f\epsilon_y]).$$

By automatically learning relevant exposure mapping function, we aim to directly minimize the error terms involving $\epsilon_e$ and the downstream error $\epsilon_y$. Other estimators (e.g., Doubly robust or orthogonal learning after handling unknown exposure mapping function) can be employed in future work for more tight error bounds.

A.5 DATASET GENERATION

For the Barabasi Albert (BA) model, the preferential attachment parameter $m \in [1, 5, 10]$ is used to generate sparse to dense networks, where a new node connects to $p_{ba}$ existing nodes to form the network. For the Watts Strogatz (WS) model, we set mean degree parameters $k \in \{0.002N, 0.005N, 0.01N\}$ with fixed rewiring probability of $0.5$, similar to prior works (Yuan et al., 2021; Adhikari and Zheleva, 2025). For the Stochastic Block Model (SBM) model, we use the number of blocks parameters $b \in \{500, 200, 100\}$ with randomly generated edge probabilities within and across communities. We also use two real-world social networks BlogCatalog and Flickr with more realistic topology and attributes to generate treatments and outcomes. We use LDA (Blei et al., 2003) to reduce the dimensionality of raw features to 50.

**Treatment model**. The treatment assignments could depend on the unit's covariates as well as peer covariates and some edge attribute. We generate treatment $T_i$ for a unit $v_i$ as $T_i \sim \theta\big(a(\tau_c \mathbf{W}_T \times \frac{\sum_{j \in \mathcal{N}_i} \mathbf{X^c}_j}{\sum_{j \in \mathcal{N}_i} Z^c_{ij}}) + (1 - \tau_c)\mathbf{W}_T \cdot \mathbf{X^c}_i\big)$, where $\theta$ denotes Bernoulli distribution, $a : \mathbb{R} \mapsto [0, 1]$ is an activation function, $\tau_c \in [0, 1]$ controls spillover influence from unit $v_i$'s peers, $\mathbf{X^c} \subset \mathbf{X}$ is a subset of node attributes, $Z^c \in \mathbf{Z}$ is an edge attribute, and $\mathbf{W}_T$ is a weight matrix.

**Outcome model**. The outcomes depend on unit's treatment, peer treatments based on the local neighborhood condition, the confounders, and the effect modifiers. We generate outcome $Y_i$ for a unit $v_i$ as:

$$Y_i = (\delta_{exp} + \delta_{em} \times T_i) \times \phi_e(G, \mathbf{X}, \mathbf{Z}, T_{-i}) + \tag{12}$$
$$(\tau_d + \tau_{em} \times \phi_{em}(G, \mathbf{X}, \mathbf{Z})) \times T_i + g(\mathbf{X_c}, Z_c, G) + \epsilon.$$

Here, the first term $(\delta_{exp} + \delta_{em} \times T_i) \times \phi_e(G, \mathbf{X}, \mathbf{Z}, T_{-i})$ captures peer effects, where $\phi_e(G, \mathbf{X}, \mathbf{Z}, T_{-i})$ captures true peer exposure that depends on local neighborhood condition (e.g., the number of mutual connections between treated peers and ego unit or attribute similarity) and $\delta_{exp}$ and $\delta_{em}$ are coefficients controlling magnitude/direction of peer effects. The term $g(\mathbf{X_c}, Z_c, G)$ captures confounding and $\epsilon \sim \mathcal{N}(0, 1)$ is random noise. The remaining term captures direct effect due to unit's own treatment with effect modification by some contexts. For semi-synthetic data, to generate heterogeneous peer effects, we use additional effect modification due to a unit's covariates, i.e., $\delta_{em} \times T_i \times \phi_v(\mathbf{X_{em}})$, where $\mathbf{X_{em}} \subset \mathbf{X}$ and $\phi_v$ is a weighted mean function with randomly generated weights. Please refer to the source code in the repository for detailed implementation of data generation. When the underlying mechanism depends on the number of mutual connections between treated peers and ego unit or attribute similarity between peers and the ego unit, the ground truth exposure mapping $\phi_e$ is defined as $\phi_e(G, \mathbf{X}, \mathbf{Z}, T_{-i}) = \frac{w_{ij}t_j}{\sum w_{ij}}$, where $w_{ij} = cardinality(\{v_k : e_{ik} \in \mathcal{E} \wedge e_{jk} \in \mathcal{E}\})$ for the former mechanism and $w_{ij} = rbf(\mathbf{x}_i, \mathbf{x}_j)$, where $rbf$ is a Gaussian radial basis function. For mechanisms involving clustering coefficient and connected components between treated peers, corresponding graph function is called in a subgraph with treated peers, i.e., $\phi_e(G, \mathbf{X}, \mathbf{Z}, T_{-i}) = graph\_func(subgraph(v_k : v_k \in \mathcal{N}_i \wedge t_k = 1))$.

A.6 ADDITIONAL EXPERIMENTAL SETTINGS

**Model implementation, hyperparameters, and model selection**. We have used $\lambda_{ent}, \lambda_{sp}, \lambda_{bal}, \lambda_{cov}, \lambda_{L1}$ as hyperparameters in our loss function. For a set of hyperparameters, we choose reasonable values and for the rest we use Python's "Ray Tune" framework for hyperparameter tuning. Although ground truth causal effects are unavailable to truly tune hyperparameters, our error analysis (extended from Shalit et al. (2021)'s work) shows that error in factual outcome prediction (and IPM distance metric) can be used as a proxy for hyperparameter tuning.

First, we describe the choice of values for regularization hyperparameters $\lambda_{ent}, \lambda_{sp}$, and $\lambda_{L1}$.

- Entropy regularization coefficient $\lambda_{ent} = 1$ to promote mask weights $\mathbf{W}_{mask} \in [0, 1]$ approaching 0 or 1 such that average entropy is low. An extremely low value does not enable the intended behavior of the soft switch for enabling or disabling certain features, and a large value could interfere with other loss terms.
- Sparsity regularization coefficient $\lambda_{sp} = 0.1$ to encourage sparse mask weights i.e., a few weights approaching 1. In conjunction with entropy loss, a value that is too high could lead all weights toward zero, and a value that is too low could produce non-sparse weights.

- L1-regularization coefficient $\lambda_{L1} = 1$ for low-dimensional synthetic data to encourage highly sparse model parameters and $\lambda_{L1} = 0.1$ for comparatively higher-dimensional semi-synthetic data to encourage sparse model parameters.

We tune the coverage parameter $\lambda_{cov} \in \{0.01, 0.1\}$ for semi-synthetic data but choose a conservative $\lambda_{cov} = 0.01$ for synthetic data for efficiency. We choose the covariate-balancing hyperparameter $\lambda_{bal} = 0.8$ based on the analysis of the original paper (Shalit et al., 2017). We set the output embedding dimension of the exposure encoder MLP to 3, giving a 6-dimensional peer exposure representation. We use $1 - layer$ deep MPGNNs for feature and exposure mapping functions. Moreover, we perform grid search hyperparameter tuning by varying GNN learning rate $\{0.1, 0.04, 0.02, 0.01\}$, and setting TARNet learning rate to $0.01$. We use Adam optimizer with weight decay of $10^{-5}$ and the learning rate is decayed by $50\%$ after 50 epochs. A 20% held-out dataset is used for model selection, where model with lowest outcome prediction loss $L_{Y_i}$ is chosen for reporting. We employ model checkpointing every other epoch to select the best performing model in a total of 100 epochs. Our implementation is similar to Adhikari and Zheleva (2025)'s INE-TARNet (also known as IDE-Net in original paper) in terms of MLP with residual network architecture, parameter tuning and model selection, and data generation.

The baselines INE-TARNet and GNN-TARNet-Motifs are also tuned similarly to our method by conducting grid search of the GNN's learning rate with $\{0.2, 0.02\}$ and variance smoothing regularization hyperparameter with $\{0.1, 1\}$, keeping TARNet's learning rate $0.02$ and other hyperparameters default. DWR is calibrated for 5 epochs to balance representation. For other baselines, we use default hyperparameters.

**Implementation of baselines**. We use publicly available code shared for the baselines INE-TARNet (Adhikari and Zheleva, 2025), TNet (Chen et al., 2024), NetEstimator (Jiang and Sun, 2022), and CauGramer (Wu et al., 2025). We adapt the code provided by authors to extend it for peer effect estimation for AEMNet (Mao et al., 2025). We implement 1GNN-HSIC (Ma and Tresp, 2021) and DWR (Zhao et al., 2024) ourselves following the paper as closely as possible. GNN-TARNet-MOTIFS is available as a baseline of INE-TARNet.

**Computational resources**. All the experiments are performed in a machine with the following resources.

- CPU: AMD EPYC 7662 64-Core Processor (128 CPUs)

- Memory: 256 GB RAM

- Operating system: Ubuntu 20.04.4 LTS

- GPU: NVIDIA RTX A5000 (24 GB)

- CUDA Version: 11.4

## A.7 SYNTHETIC DATA EXPERIMENTS AND RESULTS

Figures 6 to 10 show the performance of our method and baselines for three synthetic networks when the underlying peer exposure mechanisms depend on clustering coefficient, connected components, number of mutual connections, tie strengths, and attribute similarity. The results discussed in the main paper apply to additional peer exposure mechanisms and data generation conditions.

## A.8 SEMI-SYNTHETIC DATA EXPERIMENTS AND RESULTS

We present results for RQ2, that investigates performance in more realistic settings, for the larger benchmark network dataset Flickr in Table 4. EGONETGNN-CFR$^+$ is the best performing model in all settings. EGONETGNN-TARNet is either better than all the baselines or competitive to stable baselines INE-TARNet and GNN-Motifs. For mechanisms involving network structures, sophisticated models like CauGramer and TNet have poor performance as they are unable to generalize or converge for some data generation and model initialization settings.

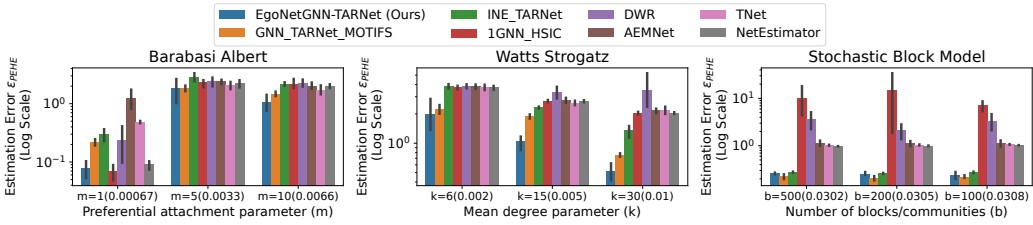

Figure 6: Peer effect estimation error when true peer exposure depends on clustering coefficient among treated peers. Our method is better than or competitive to baseline using predetermined causal network motif counts when the underlying peer exposure mechanism can be explained by causal network motif counts.

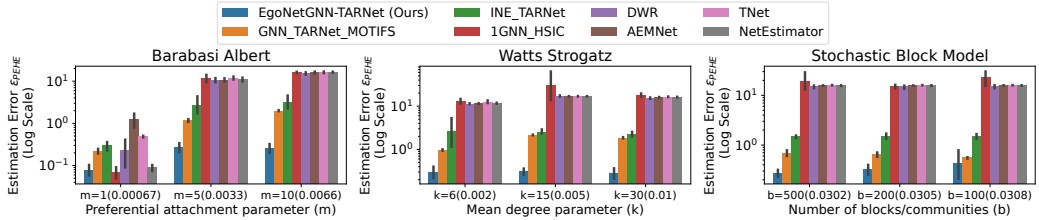

Figure 7: Peer effect estimation error when true peer exposure depends on number of mutual connections with the ego. Our method significantly outperforms all baselines showing its capability to count closed triad network motifs (i.e., triangle substructures) in the ego network.

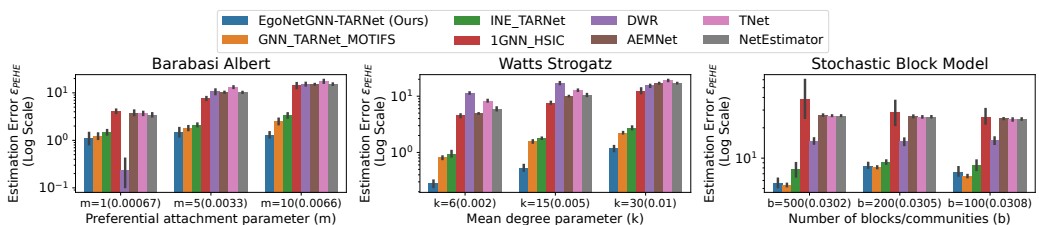

Figure 8: Peer effect estimation error when true peer exposure depends on connected components among treated peers. Our method performs well compared to all baselines when underlying peer exposure mechanism cannot be explained totally with causal network motif structures only.

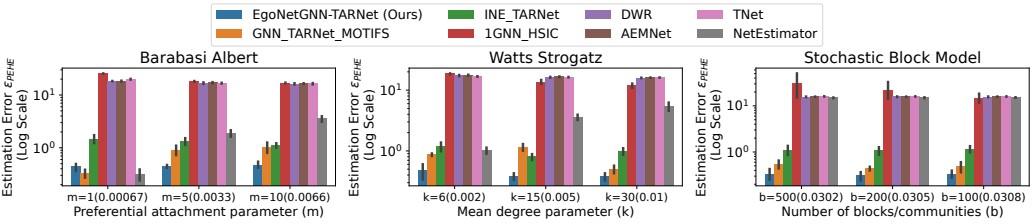

Figure 9: Peer effect estimation error when true peer exposure depends on tie strengths between ego and treated peers. Our method consistently outperforms all baselines because it can incorporate edge attributes and learn if those attributes are relevant for underlying peer exposure mechanisms.

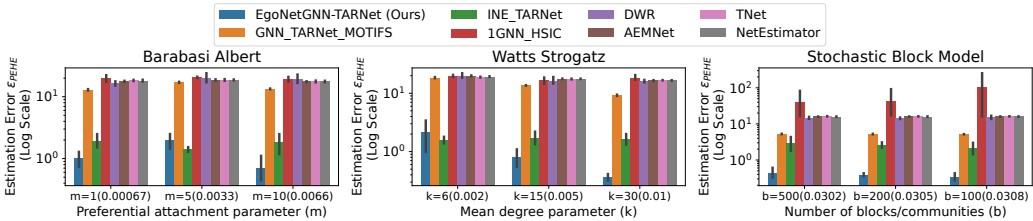

Figure 10: Peer effect estimation error when true peer exposure depends on attribute similarity between ego and treated peers. Our method consistently outperforms all baselines because it can capture and learn if attribute similarity are relevant for underlying peer exposure mechanisms.

Table 4: Mean and standard deviation (across 5 data simulations with 3 random model initializations each) of peer effect estimation error ($\epsilon_{PEHE}$) for different methods in **Flickr** dataset for four settings when true peer exposure mechanisms depend on clustering coefficients, connected components, mutual connections, and attribute similarity.

| Mechanisms | Ours-TARNet | Ours-CFR$^+$ | GNN-Motifs | INE-TARNet | 1GNN-HSIC | DWR | AEMNet | TNet | NetEst | CauGramer |
|---|---|---|---|---|---|---|---|---|---|---|
| Clus. Coef. | $7.38_{\pm 3.0}$ | $\mathbf{3.71}_{\pm 4.2}$ | $\underline{5.05}_{\pm 1.9}$ | $5.64_{\pm 0.8}$ | $13.36_{\pm 8.9}$ | $9.95_{\pm 2.8}$ | $7.45_{\pm 1.9}$ | $19.12_{\pm 19.3}$ | $10.57_{\pm 3.4}$ | $22.50_{\pm 35.0}$ |
| Con. Comp. | $\underline{1.64}_{\pm 1.1}$ | $\mathbf{1.19}_{\pm 1.3}$ | $2.34_{\pm 0.7}$ | $2.38_{\pm 0.9}$ | $19.16_{\pm 28.3}$ | $3.32_{\pm 1.0}$ | $11.59_{\pm 10.5}$ | $7.86_{\pm 7.6}$ | $3.77_{\pm 1.3}$ | $22.57_{\pm 20.4}$ |
| Mut. Con. | $2.72_{\pm 1.7}$ | $\mathbf{2.20}_{\pm 1.8}$ | $2.64_{\pm 1.1}$ | $\underline{2.41}_{\pm 1.1}$ | $3.22_{\pm 1.3}$ | $3.67_{\pm 1.6}$ | $5.72_{\pm 4.1}$ | $14.63_{\pm 24.3}$ | $3.19_{\pm 1.4}$ | $31.27_{\pm 36.5}$ |
| Attr. Sim. | $\underline{8.27}_{\pm 7.8}$ | $\mathbf{6.97}_{\pm 5.8}$ | $11.57_{\pm 12.3}$ | $10.66_{\pm 11.3}$ | $21.12_{\pm 15.4}$ | $25.29_{\pm 14.8}$ | $13.56_{\pm 12.1}$ | $40.51_{\pm 55.6}$ | $18.58_{\pm 11.7}$ | $13.70_{\pm 11.9}$ |

## A.9 ABLATION STUDIES AND HYPERPARAMETER SENSITIVITY

Table 5 shows the performance of EGONETGNN for different output dimension of peer exposure embedding $\boldsymbol{\rho}_i$ for four settings when true peer exposure mechanisms depend on clustering coefficients, connected components, mutual connections, and attribute similarity. As seen in the results, lower-dimensional peer exposure embeddings could lose expressiveness, while higher dimensions could introduce variance due to irrelevant contexts or violations of positivity. Lower-dimensional peer exposure embedding has better performance for simpler peer exposure mechanism like clustering coefficient and higher-dimensional peer exposure embedding has better performance for complex peer exposure mechanism like attribute similarity.

Table 6 shows the performance of EGONETGNN and top baselines in the BlogCatalog Data when the network is augmented to make it noisy by randomly removing or adding 10 and 20 percent of edges. We expect the models to perform worse with higher noise but we want to test whether EGONETGNN is stable or whether it degrades compared to the top baseline models. The results show that for different noisy settings, our model is consistently better than the baselines, specifically when the underlying peer exposure mechanism depends on complex structures like connected components and clustering coefficients. The results show that our model is more susceptible to noise compared to baselines for peer exposure mechanisms with attribute similarity.

## A.10 ADDITIONAL ANALYSES

**Study of variance due to data simulation and random model initializations**. We study the variation in performance of our models and prominent baselines for BlogCatalog data, where true peer exposure depends on mutual connections. We use three different random initializations of the models for the same data-generating process simulation to see the variance due to model initializations. As expected, Table 7 shows that there is low variance due to model initializations and more variance due to data simulation. This is because peer exposure resulting from some configurations of neighborhood treatment may be inherently more difficult to capture than others. In this experiment, although explicit

Table 5: Performance of EGONETGNN-CFR$^+$ in BlogCatalog Data for different output dimension of peer exposure embedding $\rho_i$ for four settings when true peer exposure mechanisms depend on clustering coefficients, connected components, mutual connections, and attribute similarity.

| Output Dimension Mechanism | 2 | 6 | 10 |
|---|---|---|---|
| Clustering Coefficient | $7.04_{\pm7.3}$ | $\mathbf{0.95}_{\pm0.5}$ | $0.97_{\pm0.7}$ |
| Connected Components | $3.24_{\pm1.8}$ | $1.50_{\pm0.7}$ | $\mathbf{1.45}_{\pm0.6}$ |
| Mutual Connections | $7.82_{\pm2.7}$ | $2.24_{\pm1.6}$ | $\mathbf{1.66}_{\pm0.7}$ |
| Attribute Similarity | $4.64_{\pm2.8}$ | $\mathbf{3.65}_{\pm2.4}$ | $3.81_{\pm2.9}$ |

Table 6: Performance of EGONETGNN and top baselines in the BlogCatalog Data when the network is augmented to make it noisy by randomly removing or adding a certain percentage of edges.

| Mechanism | Edge Augmentation Estimator | -20% | -10% | 0% | 10% | 20% |
|---|---|---|---|---|---|---|
| Attribute Similarity | EgoNetGNN-CFR+ | $4.55_{\pm2.4}$ | $4.54_{\pm3.2}$ | $3.78_{\pm2.5}$ | $2.89_{\pm1.1}$ | $4.85_{\pm3.0}$ |
| | GNN_MOTIFS | $5.40_{\pm2.9}$ | $5.77_{\pm3.7}$ | $4.53_{\pm1.9}$ | $4.82_{\pm2.0}$ | $3.79_{\pm2.9}$ |
| | INE_TARNet | $4.11_{\pm1.8}$ | $3.67_{\pm2.0}$ | $3.60_{\pm2.1}$ | $4.15_{\pm2.1}$ | $4.76_{\pm2.2}$ |
| Clustering Coefficient | EgoNetGNN-CFR+ | $1.60_{\pm0.6}$ | $1.31_{\pm0.5}$ | $1.06_{\pm0.8}$ | $2.39_{\pm2.1}$ | $1.61_{\pm0.5}$ |
| | GNN_MOTIFS | $2.56_{\pm1.2}$ | $2.17_{\pm0.8}$ | $2.34_{\pm1.1}$ | $2.27_{\pm1.0}$ | $2.59_{\pm1.1}$ |
| | INE_TARNet | $2.43_{\pm0.7}$ | $2.63_{\pm1.2}$ | $2.28_{\pm0.8}$ | $2.52_{\pm1.0}$ | $2.47_{\pm0.9}$ |
| Connected Components | EgoNetGNN-CFR+ | $3.08_{\pm0.5}$ | $2.56_{\pm0.5}$ | $1.93_{\pm1.1}$ | $2.54_{\pm0.3}$ | $3.36_{\pm1.3}$ |
| | GNN_MOTIFS | $4.14_{\pm0.6}$ | $4.34_{\pm0.9}$ | $4.96_{\pm1.6}$ | $4.89_{\pm1.3}$ | $4.74_{\pm1.5}$ |
| | INE_TARNet | $4.34_{\pm0.5}$ | $4.43_{\pm0.7}$ | $4.53_{\pm0.6}$ | $4.52_{\pm0.7}$ | $4.62_{\pm0.8}$ |
| Mutual Connections | EgoNetGNN-CFR+ | $3.06_{\pm1.7}$ | $2.09_{\pm1.5}$ | $2.14_{\pm1.7}$ | $3.06_{\pm2.1}$ | $4.78_{\pm3.3}$ |
| | GNN_MOTIFS | $3.28_{\pm1.7}$ | $3.22_{\pm1.4}$ | $2.61_{\pm0.9}$ | $2.58_{\pm1.0}$ | $3.79_{\pm2.9}$ |
| | INE_TARNet | $3.02_{\pm1.2}$ | $2.57_{\pm1.0}$ | $2.36_{\pm0.9}$ | $2.74_{\pm0.5}$ | $2.70_{\pm0.7}$ |

Table 7: Performance (in BlogCatalog Data in terms of PEHE) of our models and prominent baselines, with true peer exposure mechanism depending on mutual connections, for each simulation (i.e., data generating process seed) and three random model initializations.

| Simulation | EgoNetGNN-TARNet | EgoNetGNN-CFR$^+$ | GNN-MOTIFS | INE_TARNet | CauGramer |
|---|---|---|---|---|---|
| 0 | 2.58±0.46 | **2.23±0.55** | 4.65±1.04 | 2.41±0.34 | 5.60±2.83 |
| 1 | 1.63±1.27 | **0.72±0.31** | 2.03±0.12 | 1.98±0.06 | 3.83±0.05 |
| 2 | 3.45±0.91 | 4.06±1.21 | **3.07±0.53** | 3.16±0.23 | 5.38±2.62 |
| 3 | 3.49±0.46 | 3.51±0.78 | **3.04±0.16** | 3.68±0.49 | 5.85±2.22 |
| 4 | 3.16±2.36 | **0.70±0.10** | 1.26±0.07 | 1.25±0.03 | 5.25±1.82 |
| Overall | 2.86±1.31 | **2.24±1.55** | 2.81±1.26 | 2.50±0.92 | 5.18±1.96 |

counting of graph motifs performs better in some simulations, EgoNetGNN-CFR$^+$ exhibits the most robust performance across all simulations.

**Study of the contribution of the autoencoder variant of CFR compared to the original CFR.** Our EGONETGNN-CFR$^+$ architecture implements an autoencoder architecture with reconstruction loss in addition to IPM balance loss, as this helps mitigate the potential loss in expressiveness while balancing representations across treatment groups. To test this, we run an experiment to compare the original CFR without an autoencoder to two variants of our method and the best baseline. The results in Table 8 show our variant with the autoencoder has almost always better performance than the original CFR without an autoencoder. The balancing approach without an autoencoder seems to

Table 8: Performance (in BlogCatalog Data in terms of $\epsilon_{PEHE}$) of three variants of outcome models in our method: EgoNetGNN-TARNet (without balancing), EgoNetGNN-CFR (original CFR with balancing), and EgoNetGNN-CFR$^+$ (CFR w/ autoencoder). Experiments are conducted for five different data simulations, each with three random model initializations.

| Mechanism | EgoNetGNN-TARNet | EgoNetGNN-CFR$^+$ | EgoNetGNN-CFR (Original) | INE-TARNet |
|---|---|---|---|---|
| Clustering Coefficient | 2.13±1.88 | **0.95±0.54** | 2.03±1.56 | 2.35±0.71 |
| Structural Diversity | **1.47±0.90** | 1.50±0.68 | 1.57±0.67 | 4.78±1.09 |
| Mutual Connections | 2.86±1.31 | **2.24±1.55** | 4.32±2.32 | 2.50±0.92 |
| Attribute Similarity | 3.95±2.66 | 3.65±2.40 | 4.14±1.84 | **3.59±1.83** |

degrade the performance, even compared to our model without balancing, because the expressive representation learned by earlier modules may not be preserved.

**Study of sensitivity to balance loss coefficient**. In our experiments, we selected the covariate-balancing hyperparameter $\lambda_{bal} = 0.8$ based on the analysis presented in the original paper (Shalit et al., 2017). We analyze the sensitivity of the $\lambda_{bal}$ hyperparameter on performance. The results favor values of hyperparameter less than 1, but there is variance in the performance. Therefore, if computation resources and time are available, it is best to use hyperparameter tuning and model selection.

Table 9: Sensitivity analysis of hyperparameter $\lambda_{bal}$ in BlogCatalog Data with attribute similarity as underlying peer influence mechanism. Experiments are conducted for five different data simulations, with a fixed model initialization.

| $\lambda_{bal}$ | 0.2 | 0.4 | 0.6 | 0.8 | 1.0 |
|---|---|---|---|---|---|
| Attribute Similarity | 3.75±2.55 | **3.43±2.29** | 3.94±2.64 | 3.78±2.55 | 4.33±3.54 |

**Study of the contribution of coverage loss in model selection for robust counterfactual prediction**.

Table 10: Comparison of selection strategies using outcome-only features ($L_y$) vs. outcome + covariates ($L_y + L_{cov}$).

| Simulation | Selection ($L_y$) | Selection ($L_y + L_{cov}$) |
|---|---|---|
| 0 | 2.23±0.55 | 2.23±0.55 |
| 1 | 5.43±4.67 | 0.72±0.31 |
| 2 | 10.31±4.59 | 4.06±1.21 |
| 3 | 3.70±0.69 | 3.51±0.78 |
| 4 | 4.10±5.39 | 0.70±0.10 |

**Study of performance under homogeneous peer exposure**. We ran additional experiments to show the performance of our methods and baselines in the BlogCatalog Dataset when the true peer exposure mechanism depends on the fraction of treated peers. This is the simplest setting in which all baselines make the right exposure mapping function assumption. When the baselines know what the true mechanism is, our method still performs better than or competitive with the baselines. Interestingly, even though the peer exposure mechanism is simple, the baselines suffer heavily. To investigate this further, we remove effect modification and flipped counterfactual settings and find that these factors impact the performance of the baselines. The variants of our model are still better even in the simple setting.

**Study of empirical runtime and memory requirements for the benchmark data**. Appendix A.4 discussed the relative worst-case runtime complexity compared to approaches based on message passing GNN and an approach involving counting causal motifs. We include comparisons of rough time taken and memory usage to run experiments in the benchmark dataset BlogCatalog (5,196 nodes and 171,743 edges). We note that the implementations of our method and other baselines are not optimized for efficient memory usage, as they are designed with reproducibility and simplicity in

Table 11: Performance in BlogCatalog Dataset in terms of PEHE metric (five data simulations with one model initialization) when true peer exposure mechanism depends on the fraction of treated peers. Comparison of various models with/without effect modifications and with flipped counterfactuals or without flipped, i.e., no neighbor treated, counterfactuals.

| | Ours-TARNet | Ours-CFR⁺ | GNN-MOTIFS | INE_TARNet | 1GNN_HSIC | DWR-5 | AEMNet-CFR | TNet | NetEstimator | CauGramer |
|---|---|---|---|---|---|---|---|---|---|---|
| w/ EM | 2.40±2.74 | **2.35±1.82** | 4.25±2.22 | 4.00±1.75 | 13.90±2.80 | 16.65±1.96 | 14.36±3.81 | 13.49±2.94 | 11.76±2.20 | 8.48±6.17 |
| w/o EM | **1.05±0.93** | 3.18±3.62 | 2.17±2.70 | 2.47±2.34 | 12.80±4.37 | 15.31±2.41 | 11.82±1.82 | 12.58±4.12 | 10.58±1.34 | 4.99±1.54 |
| w/o EM & w/o flip | **0.74±0.82** | 1.90±1.90 | 2.22±2.53 | 2.07±1.66 | 1.36±1.22 | 1.54±2.03 | 3.39±2.49 | 0.83±1.26 | 2.97±1.92 | 2.96±0.88 |

mind (not production use). The results below support the theoretical analysis, where our runtime is slightly more than MPGNN-based approaches, but it is compensated for by a performance gain.

Table 12: Runtime and memory requirements for experiment in BlogCatalog Data with homogeneous exposure as underlying mechanism.

| | EgoNetGNN-TARNet | EgoNetGNN-CFR+ | GNN-MOTIFS | INE_TARNet | 1GNN_HSIC | DWR-5 | AEMNet-CFR | TNet | NetEstimator | CauGramer |
|---|---|---|---|---|---|---|---|---|---|---|
| training time (minutes) | 10.44 | 10.78 | 12.5 | 0.8 | 0.12 | 0.47 | 1.29 | 9.21 | 4.28 | 1.28 |
| GPU memory (GB) | 2.8 | 2.9 | 2.0 | 2.6 | 2.2 | 2.3 | 2.0 | 2.1 | 2.1 | 4.3 |
| $\epsilon_{PEHE}$ | 2.40±2.74 | **2.35±1.82** | 4.25±2.22 | 4.00±1.75 | 13.90±2.80 | 16.65±1.96 | 14.36±3.81 | 13.49±2.94 | 11.76±2.20 | 8.48±6.17 |

**Study of performance under censored and noisy features**. We have already performed an experiment (Table 8 in the Appendix) with noisy networks by augmenting edges (i.e, randomly adding and removing 10% and 20% edges in the overall network), and our method's performance is consistently better than the top-performing baselines. For the situation where some node features are missing, we expect the performance will be stable as long as these features are unrelated to the underlying mechanism of peer exposure or confounding. Similarly, if these missing features are correlated to other available features or network conditions, the performance is expected to have less of an impact. We performed an experiment with randomly censoring (setting to zero) 10% of the features and adding Gaussian noise $\mathcal{N}(0, 0.05)$ to the features in semi-synthetic BlogCatalog data with attribute similarity as the underlying mechanism to study the sensitivity to small measurement errors and missing features.

Table 13: Sensitivity to noisy and censored network attributes in terms of $\epsilon_{PEHE}$ metric in the BlogCatalog Data for attribute similarity as true peer exposure mechanism. Here, 10% node features are randomly set to 0 to simulate missing data and a noise is added to simulate measurement error.

| | EgoNetGNN-TARNet | EgoNetGNN-CFR+ | GNN_TARNet_MOTIFS | INE_TARNet | CauGramer |
|---|---|---|---|---|---|
| Attribute Similarity | **3.53±1.49** | 3.54±2.14 | 4.19±1.75 | 3.69±1.94 | 10.77±1.71 |

**Study of performance under violation of assumptions**. We add an experiment to test the sensitivity to violation of the confounding and interference assumption by generating data where there exists confounding and interference from two-hop neighbors. For confounding, both treatment assignments and outcome generation rely on aggregated neighbor attributes, with first-hop neighbors having a greater weight than second-hop neighbors. To model 2-hop interference, the outcome is influenced by the treatments of 2-hop neighbors, with the degree of influence depending on attribute similarity between the ego node and the neighbors. In the results below, although the magnitude of error has increased, our methods are competitive with or better than the prominent baselines.

Table 14: Heterogeneous peer effect estimation error ($\epsilon_{PEHE}$) when both confounding and interference from two-hop neighbors are present in BlogCatalog Data, when the true peer exposure mechanism depends on attribute similarity. The reported results are for 5 data simulations with one fixed model initialization seed.

| | EgoNetGNN-TARNet | EgoNetGNN-CFR+ | GNN_TARNet_MOTIFS | INE_TARNet | CauGramer |
|---|---|---|---|---|---|
| Attribute Similarity | 10.41±4.71 | **9.74±4.07** | 10.47±4.28 | 10.22±4.30 | 11.86±6.48 |

