# OpenReview forum: "Learning Exposure Mapping Functions for Inferring Heterogeneous Peer Effects"
_ICLR.cc/2026/Conference — ICLR 2026 Poster_

### Official Review · Reviewer_DGdQ · 2025-10-25

**Soundness:** 3
**Presentation:** 3
**Contribution:** 3
**Rating:** 6
**Confidence:** 3

**Summary:**

The paper proposes a new, custom GNN architecture, EgoNetGNN, for the network interference problem. The idea is to learn the underlying exposure mapping behind the network interference.  EgoNetGNN is more expressive than standard GNN architectures that have been used for this task in previous works, allowing EgoNetGNN to exhibit superior counterfactual estimation performance compared to such competing baselines, which is validated on synthetic and semi-synthetic interference settings.

**Strengths:**

Developing a novel GNN architecture that can automatically learn exposure mappings for heterogeneous peer effect estimation is a significant contribution over using out of the box GNNs. The authors also motivate this novel architecture by showing, theoretically, a gap in expressivity. The definitions and claims around these results are clearly stated. The architecture of EgoNetGNN is clearly stated as well as the training procedure. Within the given experimental settings, EgoNetGNN exhibits generally superior performance to the baseline methods, and in scenarios where that it is not the case it is explained that the baseline method is able to express the given specific exposure, while a strength of EgoNetGNN lies in expressiveness across each of eg. mutual connections, clustering coefficient, attribute similarity. This reflects EgoNetGNN being able to learn the appropriate exposure mapping in general while baseline methods may be more limited. Experimental details are presented well and ablation studies are included, strengthening the empirical results.

**Weaknesses:**

The weaknesses of the paper do not rest in whether it adequately addresses the scope that it claims to cover, but rather the significance of the scope and setting compared to previous baselines. In particular, at least as written, it felt that exposure mechanisms were chosen not because of practical relevance but because this is where the improvement compared to baselines emerges.

Even beyond the scope of GNN based approaches, as a method for estimating counterfactuals under network interference, EgoNetGNN relies on the, at least in some settings, strong assumption of known network structure and features. While reasonable for some setting such as perhaps a social media platform, this limits the scope of the work to the broader network interference problem.

**Questions:**

Could the authors provide further significance for the practical relevance of exposure mechanisms that they demonstrate superior performance over? For example, empirical works motivating the given exposure mechanism, etc.

Would it be possible to compare the performance of EgoNetGNN to the baselines under exposure mechanisms that more of the baselines actually are expressive enough to handle? Basically if practitioners were to use your method not knowing wha the true underlying exposure mapping should be, and it ends up being one that other methods could also express, what kind of relative performance might we expect?

If a practitioner wanted to EgoNetGNN but had missing graph information, would there be any recommended modifications or imputation methods?

---

> ### Author Response · Authors · 2025-11-21
> **Rebuttal Part 1**
>
> Thank you for providing the constructive comments. Below we have adressed each point in details.
>
> ### Significance and practical relevance of complex exposure mapping function learning (W1, Q1)
> > **W1.** The weaknesses of the paper do not rest in whether it adequately addresses the scope that it claims to cover, but rather the significance of the scope and setting compared to previous baselines. In particular, at least as written, it felt that exposure mechanisms were chosen not because of practical relevance but because this is where the improvement compared to baselines emerges.
>
> > **Q1.** Could the authors provide further significance for the practical relevance of exposure mechanisms that they demonstrate superior performance over? For example, empirical works motivating the given exposure mechanism, etc.
>
> Our work is motivated by the Yuan et al., 2021 paper where the authors establish causal network motifs as an automated way to capture peer exposure. They also show practical application in A/B testing in the Facebook platform. As discussed in the introduction and related work, we build upon their work to propose more flexible and expressive peer exposure representation.
>
> ### Comparisons to baseline when underlying exposure mapping is homogeneous (Q2)
>
> > **Q2.** Would it be possible to compare the performance of EgoNetGNN to the baselines under exposure mechanisms that more of the baselines actually are expressive enough to handle? Basically if practitioners were to use your method not knowing wha the true underlying exposure mapping should be, and it ends up being one that other methods could also express, what kind of relative performance might we expect?
>
> Not all baselines are capable of representing the true mechanisms. Some baselines (NetEst and TNet) can make the right assumption about the exposure mapping function only when there is simple homogeneous influence between units. Other baselines (DWR, INE-TARNet, AEMNet, CauGramer) are capable of capturing exposure mapping functions that depend on attribute similarity. The more complex ones, GNN_TARNet-MOTIFS and INE-TARNet, can handle exposure mapping function that depends on number of mutual connections and GNN_TARNet-MOTIFS is also expressive enough to capture mechanisms involving clustering coefficient among treated.
>
> We have performed additional experiments (Table R.5 below) which show the performance of our methods and baselines in the BlogCatalog Dataset when the true peer exposure mechanism depends on the fraction of treated peers. This is the simplest setting in which all baselines make the right exposure mapping function assumption.When the baselines know what the true mechanism is, our method performs better than or is competitive with the baselines. Interestingly, even though the peer exposure mechanism is simple, the baselines suffer likely due to the fact that the data generation includes effect modifications and confounding, and the peer exposure may not be sparse (like other mechanisms involving clustering coefficient).
>
> **Table R.5: Performance in BlogCatalog Dataset in terms of PEHE metric when true peer exposure mechanism depends on the fraction of treated peers. (Five data simulations with one model initialization)**
> |          Mechanism            | EgoNetGNN-TARNet   | EgoNetGNN-CFR+   | GNN_TARNet_MOTIFS   | INE_TARNet   | 1GNN_HSIC   | DWR-5      | AEMNet-CFR   | TNet       | NetEstimator   | CauGramer   |
> |:---------------------|:-------------------|:-----------------|:--------------------|:-------------|:------------|:-----------|:-------------|:-----------|:---------------|:------------|
> | Homogeneous |  *2.40±2.74*          | **2.35±1.82**        | 4.25±2.22           | 4.00±1.75    | 13.90±2.80  | 16.65±1.96 | 14.36±3.81   | 13.49±2.94 | 11.76±2.20     | 8.48±6.17   |

---

> > ### Author Response · Authors · 2025-11-21
> > **Rebuttal Part 2**
> >
> > ### Assumptions and broader applicability (W2, Q3)
> > > **W2.** Even beyond the scope of GNN-based approaches, as a method for estimating counterfactuals under network interference, EgoNetGNN relies on the, at least in some settings, strong assumption of known network structure and features. While reasonable for some settings, such as perhaps a social media platform, this limits the scope of the work to the broader network interference problem.
> >
> > It is true that our method excels in scenarios with reliable network structures and features, especially in online social networks and digital marketplaces. However, certain application areas, like social science or public health, may consist of noisy networks, censored features, and measurement errors.
> > * To study the sensitivity of noisy networks, we have already performed an experiment (Table 8 in the Appendix) with noisy networks by augmenting edges (i.e, randomly adding and removing 10% and 20% edges in the overall network), and our method's performance is consistently better than the top-performing baselines.
> > * For the situation where some node features are missing, we expect the performance will be stable as long as these features are unrelated to the underlying mechanism of peer exposure or confounding. Similarly, if these missing features are correlated to other available features or network conditions, the performance is expected to have less of an impact. We performed an experiment by randomly censoring (setting to zero) 10% of the features and adding Gaussian noise $\mathcal{N}(0, 0.05)$ to the features in semi-synthetic BlogCatalog data with attribute similarity as the underlying mechanism to study the sensitivity to small measurement errors and missing features.
> >
> > **Table R.7: Sensitivity to noisy and censored network attributes in terms of $\epsilon_{PEHE}$ metric in the BlogCatalog Data for attribute similarity as the true peer exposure mechanism. Here, 10% node features are randomly set to 0 to simulate missing data, and noise is added to simulate measurement error.**
> > |                      | EgoNetGNN-TARNet   | EgoNetGNN-CFR+   | GNN_TARNet_MOTIFS   | INE_TARNet   | CauGramer   |
> > |:---------------------|:-------------------|:-----------------|:--------------------|:-------------|:------------|
> > | Attribute Similarity | **3.53±1.49**          | ++3.54±2.14++        | 4.19±1.75           | 3.69±1.94    | 10.77±1.71  |
> >
> > > **Q3.** If a practitioner wanted to EgoNetGNN but had missing graph information, would there be any recommended modifications or imputation methods?
> >
> > The standard or existing (domain-specific) methods (e.g., [1,2]) for imputing missing node attributes and/or correcting graph edges can be applied as a preprocessing step before using EgoNetGNN. Standard methods utilize homophily to impute features (e.g., using K nearest neighbor features) or edges (feature similarity or degree smoothing). The experiments reported above hint that our method is more robust compared to baselines in a noisy setting without imputation.
> >
> > [1] Hasibi, Ramin, and Tom Michoel. "A Graph Feature Auto-Encoder for the prediction of unobserved node features on biological networks." BMC bioinformatics 22.1 (2021): 525.
> > [2] Jin, W., Ma, Y., Liu, X., Tang, X., Wang, S., & Tang, J. (2020, August). Graph structure learning for robust graph neural networks. In Proceedings of the 26th ACM SIGKDD international conference on knowledge discovery & data mining (pp. 66-74).

---

### Official Review · Reviewer_KRe6 · 2025-10-30

**Soundness:** 2
**Presentation:** 3
**Contribution:** 3
**Rating:** 6
**Confidence:** 3

**Summary:**

The paper addresses an important and underexplored problem: *learning exposure mapping functions automatically* in causal inference under interference.
Its motivation is strong, and the proposed framework is technically original and relevant to both the causal inference and graph learning communities.
The theoretical ambition (Sec. 3.3) and experimental comprehensiveness (Sec. 4) are commendable.

However, under **top-tier conference standards**, several aspects are insufficiently rigorous:

1. The causal identification relies on strong, untested assumptions.
2. Theoretical guarantees (e.g., consistency, identifiability) are not formally proven.
3. Experimental evidence conflates representational expressiveness with causal validity.
4. Interpretability and computational scalability remain weakly supported.

**Strengths:**

* **Novel problem definition:**
  The paper formalizes *exposure mapping function learning* as a distinct subproblem, bridging a gap in causal inference with interference (Sec. 1–2).

* **Architectural innovation:**
  EGONETGNN combines ego-network extraction, masked weighting, and exposure encoding, with coverage-based losses designed for invariant and expressive exposure learning (Sec. 3.1–3.2).

* **Comprehensive experiments:**
  The study spans multiple network types and exposure mechanisms, demonstrating robustness across topologies and influence structures.

* **Theoretical ambition:**
  Expressiveness analysis (Sec. 3.3; Appendix A.4) connects to prior results on GNN motif-counting power (Chen et al., 2020) and causal motifs (Yuan et al., 2021).

* **Transparent ablations:**
  Ablation studies (Table 2) clearly show that the feature encoder and masked weights contribute meaningfully to performance.

* **Reproducibility commitment:**
  Code release and detailed configuration documentation (Appendix A.6) facilitate replication.

**Weaknesses:**

1. **Strong and untested causal assumptions.**
   The identification (Prop. 1) depends on *unconfoundedness* and *neighborhood interference* (Assumptions 2–3). These are unrealistic in most observational networks and are not empirically validated.

2. **Lack of formal theoretical guarantees.**
   The expressiveness argument shows motif-counting capability but does not establish *consistency* or *generalization bounds* for ( \hat{\phi}_e ) or ( \hat{f}_y ). No identifiability result ensures that learned exposures converge to true exposures.

3. **Unclear causal vs. representational validity.**
   Reduced εPEHE may result from representation fitting rather than correct causal identification. There are no counterfactual or ablation diagnostics that isolate causal correctness.

4. **Limited interpretability.**
   The learned exposure embeddings ( \rho ) lack transparency. Table 3 reports correlations but provides no qualitative insight into which node/edge structures the model captures.

5. **Scalability not demonstrated.**
   EGONETGNN’s per-node ego-network extraction and masked MLP operations increase complexity by roughly ( \rho_E \times \text{avg}(d) ) over MPGNNs (Sec. 3.3), yet no runtime or memory benchmarks are reported.

6. **Statistical reliability and variance.**
   The paper reports mean εPEHE but omits confidence intervals, variance decomposition, or sensitivity to hyperparameters ( \lambda_{\text{cov}}, \lambda_{\text{ent}}, \lambda_{\text{sp}} ).

7. **Incomplete comparison to modern baselines.**
   Proximal and double-negative-control causal estimators (Tchetgen et al., 2020; Miao et al., 2024) are cited but not evaluated. Transformer-based causal GNNs are missing.

8. **Potential endogeneity in joint learning.**
   Because ( \phi_e ) is optimized via outcome loss ( y ), exposure representations may absorb post-treatment bias. No regularization or identification constraint ensures causal directionality.

**Questions:**

1. **On causal identification and assumptions**

   * You rely on *unconfoundedness* and *neighborhood interference* (Assumptions 2–3) for identifiability.
     Could you clarify **how these assumptions are justified or tested** in your semi-synthetic datasets?
     For instance, are confounders explicitly simulated and controlled?
     If not, how do you ensure that violation of these assumptions does not bias the estimated heterogeneous peer effects (HPE)?
   * Have you considered or tested *partial interference* (interference beyond immediate neighbors)? If so, could you show how EGONETGNN’s performance changes under that violation?

2. **On the learned exposure mapping function ( \hat{\phi}_e )**

   * Can you provide **interpretability or visualization** of the learned peer exposure embeddings ( \rho )?
     For example, do certain dimensions correspond to structural motifs (e.g., triangles, degree centrality, clustering coefficients) or node attributes?
   * How do you ensure that ( \hat{\phi}_e ) captures *causal influence* rather than merely *correlated structural patterns*?
     Have you tried using random edge rewiring or attribute permutation tests to probe causal validity?

3. **On theoretical soundness**

   * The paper claims improved expressiveness compared to standard MPGNNs (Sec. 3.3). Could you provide **formal statements or empirical ablations** verifying that EGONETGNN can indeed represent specific causal network motifs (e.g., closed triads) that GCN/GIN cannot?
   * Is there any **consistency guarantee** that the learned exposure mapping converges to the true one as sample size grows? If not, can you at least discuss conditions under which your end-to-end loss (Eq. 11) provides an unbiased estimator of the HPE?

4. **On training stability and loss design**

   * The overall loss combines several terms (Eq. 11). How sensitive are results to the hyperparameters ( \lambda_{\text{cov}}, \lambda_{\text{ent}}, \lambda_{\text{sp}} )?
     Could you include a sensitivity plot or table showing their impact on εPEHE or embedding coverage?
   * How does the *coverage loss* quantitatively affect the learned exposure distribution—does it prevent mode collapse or just rescale embeddings?

---

> ### Author Response · Authors · 2025-11-21
> **Rebuttal Part 1**
>
> Thank you for your detailed comments. We have provided point-wise responses below.
>
> ### Assumptions (W1, Q1, Q2)
> > **W1**. Strong and untested causal assumptions. The identification (Prop. 1) depends on unconfoundedness and neighborhood interference (Assumptions 2–3). These are unrealistic in most observational networks and are not empirically validated.
>
> > **Q1**. On causal identification and assumptions
> You rely on unconfoundedness and neighborhood interference (Assumptions 2–3) for identifiability. Could you clarify how these assumptions are justified or tested in your semi-synthetic datasets? For instance, are confounders explicitly simulated and controlled? If not, how do you ensure that violation of these assumptions does not bias the estimated heterogeneous peer effects (HPE)?
>
> > **Q2**. Have you considered or tested partial interference (interference beyond immediate neighbors)? If so, could you show how EGONETGNN’s performance changes under that violation?
>
> Indeed, the need to assume unconfoundedness is a known limitation of works using observational data. In the setting of unknown peer influence mechanisms, unconfoundedness is needed even for experimental data because we could randomize (or intervene on) treatments but not directly on peer exposure. Like other works focusing on representation learning, we make unconfoundedness assumption. In Appendix A.1 we discuss its plausibility and alternative methods like using proxies, natural experiments, or instrumental variables for handling unobserved confounding. Neighborhood interference is another commonly used simplifying assumption and we have discussed its plausibility and possible extensions of considering K-hop neighbors in Appendix A.1. Given our work is one of the earliest to consider learning exposure mapping function, we limit the scope to the settings where identifiabiity is satisfied and we focus on representation learning of complex mechanisms within 1-hop neighbors.
>
> In our experiments, we simulate confounding due to a subset of unit's attributes and aggregated neighbor attributes which capture the observed attributed network. For example, a subset of node attributes and their neighborhood aggregation are simulated as confounders affecting both treatment assignments and outcome.
> If unconfoundedness is assumed but violated, then the results would have a confounding bias. The violation of neighborhood interference with influence from higher-order neighbors could cause misspecification of peer exposure also leading to biased estimates. We add an experiment to test this sensitivity by generating data where there exists confounding and interference from two hop neighbors. For confounding, both treatment assignments and outcome generation rely on aggregated neighbor attributes, with first-hop neighbors having a greater weight than second-hop neighbors. To model 2-hop interference, outcome is influenced by the treatments of 2-hop neighbors with degree of influence depending on attribute similarity between the ego node and the neighbors. Although the magnitude of error has increased, our methods are still competitive to or better than the prominent baselines.
>
>
>
> **Table R.8: Heterogenous peer effect estimation error ($\epsilon_{PEHE}$) when both confounding and interference from two hop neighbors is present in BlogCatalog Data when true peer exposure mechanism depends on attribute similarity. The reported results are for 5 data simulations with one fixed model initialization seed.**
> |                      | EgoNetGNN-TARNet   | EgoNetGNN-CFR+   | GNN_TARNet_MOTIFS   | INE_TARNet   | CauGramer   |
> |:---------------------|:-------------------|:-----------------|:--------------------|:-------------|:------------|
> | Attribute Similarity | 10.41±4.71         | **9.74±4.07**        | 10.47±4.28          | *10.22±4.30*   | 11.86±6.48  |

---

> > ### Author Response · Authors · 2025-11-21
> > **Rebuttal Part 2**
> >
> > ### Theoretical generalization bounds, asymptotic consistency, convergence, and variance (W2, W6, Q6)
> >
> > > **W2**. Lack of formal theoretical guarantees. The expressiveness argument shows motif-counting capability but does not establish consistency or generalization bounds for ( \hat{\phi}_e ) or ( \hat{f}_y ). No identifiability result ensures that learned exposures converge to true exposures.
> >
> > > **W6.** Statistical reliability and variance. The paper reports mean εPEHE but omits confidence intervals, variance decomposition ...
> >
> > > **Q6.** Is there any consistency guarantee that the learned exposure mapping converges to the true one as sample size grows? If not, can you at least discuss conditions under which your end-to-end loss (Eq. 11) provides an unbiased estimator of the HPE?
> >
> > The reviewer has highlighted the current limitations of estimators utilizing complex GNNs or Transformers for heterogeneous peer effect estimation under unknown influence mechanisms. While complex GNN and transformer architectures are very powerful in capturing confounding variables, effect modifications, and unknown influence mechanisms in complex network data, they lack interpretability and theoretical guarantees of consistency or convergence. As discussed in our paper, theoretical properties of complex GNNs have been shown in simpler settings where there is a homogeneous exposure mapping function and for average treatment effects (Khatami et al., 2024 and Chen et al. 2024). Theoretical guarantees in the setting with unknown exposure mapping functions and heterogeneous effects is an important future direction but it is outside the scope of our paper. Instead, our main theoretical results in this paper are on the expressiveness of the GNN to capture complex underlying influence mechanisms.
> > In this work, we focus on point estimates without confidence intervals. For real-world data, techniques like bootstrapping and random model initializations could be used to obtain a measure of uncertainity similar to confidence intervals. Like ours, other works have used point estimates to present empirical evidence to support robustness in complex interference settings, such as using GNN on hypergraphs to model group interactions (Ma et al., KDD 2022) and using Graph Transformers to model unknown interference structure (e.g., CauGramer (Wu et al., ICLR 2025)), leaving theoretical results on error bounds with complex models to future work. We hope our work could spur future research directions and collaborations to address these limitations. Our framework is flexible enough to be adapted to utilize unbiased estimators like Horvitz-Thompson (HT) estimator or Targeted Maximum Likelihood Estimation (TMLE) estimator for causal estimation.
> >
> > ### Endogeneity and Baselines (W8, W7)
> > > **W8.** Potential endogeneity in joint learning. Because ( \phi_e ) is optimized via outcome loss ( y ), exposure representations may absorb post-treatment bias. No regularization or identification constraint ensures causal directionality.
> >
> >
> > Assumption 1 considers that the network G along with node attributes $\mathbf{X}$ and edge attributes $\mathbf{Z}$ are measured before treatment assignments $\mathbf{t} = \pi$ and treatments are not mutable. This ensures there is causal directionality/topological ordering, i.e., G(V, E, **X**,**Z**),**t**, **p**, **y**.
> >
> >
> > >**W7.** Incomplete comparison to modern baselines. Proximal and double-negative-control causal estimators (Tchetgen et al., 2020; Miao et al., 2024) are cited but not evaluated. Transformer-based causal GNNs are missing.
> >
> > The methods using proximal and double-negative-control causal estimators require additional assumptions about which variables can serve as proxies/negative controls and are thus not straightforward to use as baselines. We do compare our method against CauGramer which is a transformer-based causal GNN.

---

> > > ### Author Response · Authors · 2025-11-21
> > > **Rebuttal Part 3**
> > >
> > > ### Training stability, hyperparameters, and loss functions (W6, Q7, Q8 )
> > > > **W6***: Statistical reliability and variance. The paper reports mean εPEHE but omits [...] sensitivity to hyperparameters ( \lambda_{\text{cov}}, \lambda_{\text{ent}}, \lambda_{\text{sp}} ).
> > >
> > > > **Q7** The overall loss combines several terms (Eq. 11). How sensitive are results to the hyperparameters ( \lambda_{\text{cov}}, \lambda_{\text{ent}}, \lambda_{\text{sp}} )? Could you include a sensitivity plot or table showing their impact on εPEHE or embedding coverage?
> > >
> > > We have used $\lambda_{ent}, \lambda_{sp}, \lambda_{bal}, \lambda_{cov}, \lambda_{L1}$ as hyperparameters in our loss function. For a set of hyperparameters, we choose reasonable values and for the rest we use Python's "Ray Tune" framework for hyperparameter tuning. Although ground truth causal effects are unavailable during training to truly tune hyperparameters, our error analysis (extended from Shalit et al. (2021)'s work) shows that error in factual outcome prediction and IPM distance metric can be used as proxies for hyperparameter tuning.
> > > First, we describe the choice of values for regularization hyperparameters $\lambda_{ent}, \lambda_{sp},$ and $\lambda_{L1}$.
> > >
> > > - Entropy regularization coefficeint $\lambda_{ent}=1$ to promote mask weights $\mathbf{W}_{mask} \in [0,1]$ approaching 0 or 1 such that average entropy is low. An extremely low value does not enable the intended behavior of soft switch for enabling or disabling certain features and a large value could interfere with other loss terms.
> > > - Sparsity regularization coefficient $\lambda_{sp}=0.1$ to encourage sparse mask weights i.e., a few weights approaching 1. In conjuction with entropy loss, a value that is too high could lead all weights toward zero and a value that is too low could produce non-sparse weights.
> > > - L1-regularization coefficient $\lambda_{L1}=1$ for low-dimensional synthetic data to encourage highly sparse model parameters and $\lambda_{L1}=0.1$ for comparatively higher-dimensional semi-synthetic data to encourage sparse model parameters.
> > >
> > > We tune the coverage parameter $\lambda_{cov} \in \\{0.01, 0.1\\}$ for semi-synthetic data but choose a conservative $\lambda_{col}=0.01$ for synthetic data for effciency. For hyperparameter tuning/model selection, we vary learning rates along with other hyperparameters and, using 20% data for validation, select the model with lowest prediction loss for synthetic data and lowest sum of prediction and coverage loss for semi-synthetic data. We choose the covariate-balancing hyperparameter $\lambda_{bal}=0.8$ based on the analysis of the original paper (Shalit et al., 2017). We analyze the sensitivity of two important hyperparameters $\lambda_{bal}$ (new experiment in Table R.2 below) and $\lambda_{cov}$ in the Appendix (Table 6 in paper), and will add the above details in the paper. We will add the analysis for other mechanisms before the end of the discussion period.
> > >
> > > **Table R.2: Sensitivity analysis of hyperparameter $\lambda_{bal}$ in BlogCatalog Data with attribute similarity as underlying peer influence mechanism.**
> > > |     $\lambda_{bal}$       | 0.2     | 0.4     | 0.6     | 0.8     | 1.0     |
> > > |:---------------------|:----------|:----------|:----------|:----------|:----------|
> > > | Attribute Similarity | ++3.75±2.55++ | **3.43±2.29** | 3.94±2.64 | 3.78±2.55 | 4.33±3.54 |
> > >
> > >
> > > > **Q8.** How does the coverage loss quantitatively affect the learned exposure distribution—does it prevent mode collapse or just rescale embeddings?
> > >
> > > The coverage loss aims to prevent the equivalent of mode collapse where the distribution of output peer exposure representation is limited. For example, if the peer exposure is low or has low variance, the model may learn to output zero or constant value, which affects counterfactual prediction and peer effect estimation. We include an experiment with two model selection strategies 1) Lowest prediction loss on validation set and 2) Lowest prediction loss + coverage loss on validation set, and show how utilizing coverage loss for model selection supports robust estimation mitigating mode collapse. The results show that model selection strategy considering coverage loss favors models without mode collapse and robust counterfactual prediction (hence reduced PEHE).
> > >
> > > **Table R.8: Using coverage loss for model selection helps in robust effect estimation. PEHE metric for EgoNetGNN-CFR+ in BlogCatalog Data with Mutual Connections as underlying peer influence mechanism. We conduct five data simulations, each with three different random model initializations.**
> > >
> > > | Simulation | Selection ($L_y$) | Selection($L_y+L_{cov}$) |
> > > | -------- | -------- | -------- |
> > > |      0 | 2.23±0.55           | 2.23±0.55              |
> > > |      1 | 5.43±4.67           | 0.72±0.31              |
> > > |      2 | 10.31±4.59          | 4.06±1.21              |
> > > |      3 | 3.70±0.69           | 3.51±0.78              |
> > > |      4 | 4.10±5.39           | 0.70±0.10              |

---

> ### Author Response · Authors · 2025-11-21
> **Rebuttal Part 4**
>
> ## Expressiveness and Interpretability (W3, W4,  Q4, Q5)
> > **W3**. Unclear causal vs. representational validity. ...
>
> > **Q4.** How do you ensure that ( \hat{\phi}_e ) captures causal influence rather than merely correlated structural patterns? Have you tried using random edge rewiring or attribute permutation tests to probe causal validity?
>
> It is possible that learned peer exposure embeddings optimized via factual prediction loss could capture a correlated mechanism instead of true mechanism. For example, in a network with homophily (neighbors with similar attributes), insteading of learning the true mechanisms like attribute similarity, it could learn homogeneous exposure like fraction of treated friends. However, if the model does not learn true mechanism, the counterfactual prediction will be biased giving relatively large error $ε_{PEHE}$. We use a setting of 100% flipped neighbor treatments as counterfactuals in different experiment settings to check the robustness of our models. To mitigate the issue where $\hat{\phi}_e$ captures a correlated pattern rather than underlying mechanism, we perform model selection by using 20% validation set based on prediction loss and coverage loss in the dataset. The idea is that choosing a correlated pattern rather than true one is akin to overfitting. Table R.8 above shows this approach can mitigate the issue due to correlated non-causal patterns.
>
> > **Q4.** Limited interpretability. The learned exposure embeddings ( \rho ) lack transparency. Table 3 reports correlations but provides no qualitative insight into which node/edge structures the model captures. Can you provide interpretability or visualization of the learned peer exposure embeddings ( \rho )? For example, do certain dimensions correspond to structural motifs (e.g., triangles, degree centrality, clustering coefficients) or node attributes?
>
> The learned peer exposure embeddings $\rho$ capture the influence from peers considering the underlying influence mechanisms. Although our work focuses on ensuring that the representation is expressive enough to capture the underlying mechanism and does not explicitly handle interpretability, the mask weights and the intermediate input to the mask weight layer could be utilized to understand the underlying influence mechanism. The embeddings from the intermediate hidden representations capture all structural motifs (Eq. 5), whereas the mask weights (Eq. 6) capture the relevant ones, the exposure encoder (Eq. 7) combines features to learn more complex ones, and finally the graph readout summarizes the low dimensional peer exposure representation. Utilizing intermediate embeddings (Eqs. 5 and 6), we can determine whether structures (first half of embeddings) or attributes (second half) are important (non-zero after masking). One could potentially utilize existing work on GNN interpretability (e.g. [1,2]) to understand the underlying mechanism which we leave to future work.
>
> [1] Armgaan, Burouj, Manthan Dalmia, Sourav Medya, and Sayan Ranu. "GraphTrail: Translating GNN predictions into human-interpretable logical rules." Advances in Neural Information Processing Systems 37 (2024)
>
> [2] Raut, Riddhiman, Romit Maulik, and Shivam Barwey. "FIGNN: Feature-Specific Interpretability for Graph Neural Network Surrogate Models." arXiv preprint arXiv:2506.11398 (2025).
>
> >**Q5.** The paper claims improved expressiveness compared to standard MPGNNs (Sec. 3.3). Could you provide formal statements or empirical ablations verifying that EGONETGNN can indeed represent specific causal network motifs (e.g., closed triads) that GCN/GIN cannot?
>
> We have added the following formal statements to the main paper (which we will update fully before discussion period ends).
> **Proposition 2 (Expressiveness of EGONETGNN)**. EGONETGNN is expressive enough to capture all dyad, open triad, closed triad, and open tetrad causal network motifs.
>
> We sketch the proof by dividing the statement into following two claims. The details of the proofs of these claims are in Appendix A.4.
> *Claim 1. EGONETGNN is as expressive as standard MPGNN in capturing dyad, open triad, and open tetrad causal network motifs.*
> *Claim 2. EGONETGNN also captures closed triad causal network motifs.*
>
> Our experimental results with clustering coefficient as underlying mechanism shows our method EgoNetGNN-CFR+ is able to capture closed triads while GCN-CFR and GIN-CFR utilizing homogeneous exposure are not.
>
> **Table R.9: Performance of our method in comparison to vanilla MPGNNs (GCN and GIN) using the fraction of treated peers as peer exposure when the underlying exposure depends on the clustering coefficient between treated peers in the BlogCatalog data. Experiments are conducted for five different data simulations, each with three random model initializations.**
> |   | EgoNetGNN-CFR+    | GCN-CFR   | GIN-CFR   |
> |:--|:--|:-|:-|
> | Clustering Coefficient | **0.95±0.54** | 2.58±0.88 | 2.39±0.69 |

---

> > ### Author Response · Authors · 2025-11-21
> > **Rebuttal Part 5**
> >
> > ## Scalability
> > > **W5.** Scalability not demonstrated. EGONETGNN’s per-node ego-network extraction and masked MLP operations increase complexity by roughly ( \rho_E \times \text{avg}(d) ) over MPGNNs (Sec. 3.3), yet no runtime or memory benchmarks are reported.
> >
> >
> > Appendix A.4 discussed relative runtime while comparing to approaches based on message passing GNN and approach involving counting causal motifs. We include comparisions of rough time taken and memory usage to run experiments in the benchmark dataset BlogCatalog (5,196 nodes and 171,743 edges). We note that the implementation of our method and other baselines are not optimized for efficient memory usage as they are designed with reproducibility and simplicity in mind (not production use).
> >
> >
> > **Table R.6: Runtime and memory requirements for experiment in BlogCatalog Data with homogeneous exposure as underlying mechanism.**
> >
> > |            |   EgoNetGNN-TARNet |   EgoNetGNN-CFR+ |   GNN-MOTIFS |   INE_TARNet |   1GNN_HSIC |   DWR-5 |   AEMNet-CFR |   TNet |   NetEstimator |   CauGramer |
> > |:-----------|-------------------:|-----------------:|--------------------:|-------------:|------------:|--------:|-------------:|-------:|---------------:|------------:|
> > | training time (minutes) |              10.44 |            10.78 |                12.5 |          0.8 |        0.12 |    0.47 |         1.29 |   9.21 |           4.28 |        1.28 |
> > | GPU memory (GB) |              2.8  |            2.9  |                2.0 |          2.6 |        2.2 |    2.3 |         2.0 |   2.1 |           2.1 |        4.3 |
> > | $\epsilon_{PEHE}$ | ++2.40±2.74++          | **2.35±1.82**        | 4.25±2.22           | 4.00±1.75    | 13.90±2.80  | 16.65±1.96 | 14.36±3.81   | 13.49±2.94 | 11.76±2.20     | 8.48±6.17   |

---

### Official Review · Reviewer_EFAX · 2025-10-30

**Soundness:** 2
**Presentation:** 3
**Contribution:** 2
**Rating:** 4
**Confidence:** 4

**Summary:**

This paper tackles the problem of estimating heterogeneous peer effects under network interference when the true exposure mapping function is unknown or misspecified.
The authors propose EGONETGNN, a graph neural network framework that learns the exposure mapping function automatically from ego-net structures and contextual features.
Comprehensive experiments on synthetic and semi-synthetic datasets demonstrate the effectiveness of the proposed method.

**Strengths:**

1. The experiments are comprehensive.
2. The paper provides the source code.

**Weaknesses:**

1. The framework still relies on the unconfoundedness and neighborhood-interference assumptions (Assumptions 2–3), which may limit its causal interpretability in real-world settings. If these assumptions are violated, the model’s learned exposure representations and effect estimates could become biased, as the identification no longer holds.

2. The abstract and introduction emphasize theory as part of the contribution, but the main text does not explicitly present any novel theoretical results.
3. Though scalability is discussed (Appendix A.4), the runtime grows with ego-network size, and large-graph applicability remains an open concern.
4. The paper only validates the model on synthetic and semi-synthetic datasets, without real-world intervention data.
5. Appendix A.4 provides runtime measurements only for EGONETGNN itself but does not include baseline comparisons. A relative runtime or scalability comparison with the baseline methods would make the analysis more complete.

**Questions:**

1. What would happen if the unconfoundedness or neighborhood-interference assumptions (Assumptions 2–3) are violated? In particular, the neighborhood-interference assumption restricts interference to first-order neighbors, which may not hold in real-world networks. How sensitive is the method to such violations?

2. The model is computationally about ρε​×avg(d) times more expensive than a standard MPGNN, which may limit its scalability to large networks.
Regarding scalability, the authors mention that sampling or parallelization could alleviate the computational burden. Could the paper provide quantitative comparisons of runtime and memory usage?

3. How does EGONETGNN perform in noisy networks or when some node features are missing? Does it still outperform the baselines under such imperfect conditions?

4. What are the key differences and advantages of EGONETGNN compared to recent methods such as AEMNet (ICASSP 2025) and CauGramer (ICLR 2025)? Could the framework potentially be extended to integrate attention or transformer-based mechanisms?

---

> ### Author Response · Authors · 2025-11-21
> **Rebuttal Part 1**
>
> Thank you for providing constructive comments. We have addressed each point in detail below.
>
> ### Assumptions (W1 and Q1)
> >**W1:** The framework still relies on the unconfoundedness and neighborhood-interference assumptions (Assumptions 2–3), which may limit its causal interpretability in real-world settings. If these assumptions are violated, the model’s learned exposure representations and effect estimates could become biased, as the identification no longer holds.
> >> **Q1:** What would happen if the unconfoundedness or neighborhood-interference assumptions (Assumptions 2–3) are violated? In particular, the neighborhood-interference assumption restricts interference to first-order neighbors, which may not hold in real-world networks. How sensitive is the method to such violations?
>
> Indeed, the need to assume unconfoundedness is a known limitation of works using observational data. Considering the setting of unknown peer influence mechanisms, unconfoundedness is needed even for experimental data because we could randomize (or intervene on) treatments but not directly on peer exposure. Like other works focusing on representational learning, we make the unconfoundedness assumption. In Appendix A.1, we discuss its plausibility and alternative methods like using proxies, natural experiments, or instrumental variables for handling unobserved confounding. Neighborhood interference is another commonly used simplifying assumption and we have discussed its plausibility and possible extensions of considering K-hop neighbors in Appendix A.1. Given our work is one of the earliest to consider learning exposure mapping function, we limit the scope to the settings where identifiability is satisfied and we focus on representation learning of complex mechanisms within 1-hop neighbors.
>
> If unconfoundedness is assumed but violated, then the results would have a confounding bias. The violation of neighborhood interference with influence from higher-order neighbors could cause misspecification of peer exposure also leading to biased estimates. We add an experiment to test this sensitivity by generating data where there exists confounding and interference from two hop neighbors. For confounding, both treatment assignments and outcome generation rely on aggregated neighbor attributes, with first-hop neighbors having a greater weight than second-hop neighbors. To model 2-hop interference, outcome is influenced by the treatments of 2-hop neighbors with degree of influence depending on attribute similarity between the ego node and the neighbors. Although the magnitude of error has increased, our methods are competitive to or better than the prominent baselines.
>
> **Table R.5: Heterogeneous peer effect estimation error ($\epsilon_{PEHE}$) when both confounding and interference from two hop neighbors is present in BlogCatalog Data when true peer exposure mechanism depends on attribute similarity. The reported results are for 5 data simulations with one fixed model initialization seed.**
> |  | EgoNetGNN-TARNet   | EgoNetGNN-CFR+   | GNN_MOTIFS   | INE_TARNet   | CauGramer   |
> |:--|:--|:-|:--|:--|:-|
> | Attribute Similarity | 10.41±4.71 | **9.74±4.07**  | 10.47±4.28   | *10.22±4.30*  | 11.86±6.48  |
>
> ### Scalability (W3, W5, and Q2)
>
> >**W3:** Though scalability is discussed (Appendix A.4), the runtime grows with ego-network size, and large-graph applicability remains an open concern.
>
> Similar to our baselines, the goal of our work is to propose a new method and evaluating it on benchmark datasets. Ensuring scalability is indeed an important future direction [1,2,3,4]. We have discussed some potential ideas in Appendix A.1. For our setting, a few ways to address large runtime and/or memory usage could be sampling ego networks to reduce the training set or sampling the neighborhood (inspired by [4]) within a K-hop ego network. From an implementation point of view, we can parallelize our framework by processing batches (similar in [2]) of ego networks in parallel exploiting modern deep learning frameworks. Please note that our current method has efficiency similar to the competitive baselines (please see the response to your next comment) while being better in quality. Moreover, our method is theoretically grounded.
>
> [1] Fey, Matthias, Jan E. Lenssen, Frank Weichert, and Jure Leskovec. "Gnnautoscale: Scalable and expressive graph neural networks via historical embeddings." ICML'21
>
>  [2] Wan, Xinchen, Kaiqiang Xu, Xudong Liao, Yilun Jin, Kai Chen, and Xin Jin. "Scalable and efficient full-graph gnn training for large graphs." Proceedings of the ACM on Management of Data 1, no. 2 (2023): 1-23
>
> [3] Gupta, Vipul, Xin Chen, Ruoyun Huang, Fanlong Meng, Jianjun Chen, and Yujun Yan. "GraphScale: A Framework to Enable Machine Learning over Billion-node Graphs." CIKM'24.
>
> [4] Serafini, Marco, and Hui Guan. "Scalable graph neural network training: The case for sampling." ACM SIGOPS Operating Systems Review 55, no. 1 (2021): 68-76.

---

> > ### Author Response · Authors · 2025-11-21
> > **Rebuttal Part 2**
> >
> > > **W5:** Appendix A.4 provides runtime measurements only for EGONETGNN itself but does not include baseline comparisons. A relative runtime or scalability comparison with the baseline methods would make the analysis more complete.
> > >> **Q2**: The model is computationally about ρε×avg(d) times more expensive than a standard MPGNN, which may limit its scalability to large networks. Regarding scalability, the authors mention that sampling or parallelization could alleviate the computational burden. Could the paper provide quantitative comparisons of runtime and memory usage?
> >
> >
> > Appendix A.4 discussed relative worst-case runtime complexity comparing to approaches based on message passing GNN and approach involving counting causal motifs. We include comparisions of rough time taken and memory usage to run experiments in the benchmark dataset BlogCatalog (5,196 nodes and 171,743 edges). We note that the implementation of our method and other baselines are not optimized for efficient memory usage as they are designed with reproducibility and simplicity in mind (not production use).
> >
> >
> > **Table R.6: Runtime and memory requirements for experiment in BlogCatalog Data with homogeneous exposure as underlying mechanism.**
> >
> > |            |   EgoNetGNN-TARNet |   EgoNetGNN-CFR+ |   GNN-MOTIFS |   INE_TARNet |   1GNN_HSIC |   DWR-5 |   AEMNet-CFR |   TNet |   NetEstimator |   CauGramer |
> > |:-----------|-------------------:|-----------------:|--------------------:|-------------:|------------:|--------:|-------------:|-------:|---------------:|------------:|
> > | training time (minutes) |              10.44 |            10.78 |                12.5 |          0.8 |        0.12 |    0.47 |         1.29 |   9.21 |           4.28 |        1.28 |
> > | GPU memory (GB) |              2.8  |            2.9  |                2.0 |          2.6 |        2.2 |    2.3 |         2.0 |   2.1 |           2.1 |        4.3 |
> > | $\epsilon_{PEHE}$ | *2.40±2.74*          | **2.35±1.82**        | 4.25±2.22           | 4.00±1.75    | 13.90±2.80  | 16.65±1.96 | 14.36±3.81   | 13.49±2.94 | 11.76±2.20     | 8.48±6.17   |
> >
> > ### Noisy network and missing data (Q3)
> > > **Q3.** How does EGONETGNN perform in noisy networks or when some node features are missing? Does it still outperform the baselines under such imperfect conditions?
> >
> > We have already performed an experiment (Table 8 in the Appendix) with noisy networks by augmenting edges (i.e, randomly adding and removing 10% and 20% edges in overall network), and our method's performance is consistently better than the top performing baselines. For the situation where some node features are missing, we expect the performance will be stable as long as these features are unrelated to underlying mechanism of peer exposure or confounding. Similarly, if these missing features are correlated to other available features or network conditions, the performance is expected to have less of an impact. We performed an experiment with randomly censoring (setting to zero) 10% of the features and adding Gaussian noise $\mathcal{N}(0, 0.05)$ to the features in semi-synthetic BlogCatalog data with attribute similarity as the underlying mechanism to study the sensitivity to small measurement errors and missing features.
> >
> > **Table R.7: Sensitivity to noisy and censored network attributes in terms of $\epsilon_{PEHE}$ metric in the BlogCatalog Data for attribute similarity as true peer exposure mechanism. Here, 10% node features are randomly set to 0 to simulate missing data and a noise is added to simulate measurement error.**
> > |                      | EgoNetGNN-TARNet   | EgoNetGNN-CFR+   | GNN_MOTIFS   | INE_TARNet   | CauGramer   |
> > |:---------------------|:-------------------|:-----------------|:--------------------|:-------------|:------------|
> > | Attribute Similarity | **3.53±1.49**          | *3.54±2.14*        | 4.19±1.75           | 3.69±1.94    | 10.77±1.71  |

---

> > > ### Author Response · Authors · 2025-11-21
> > > **Rebuttal Part 3**
> > >
> > > ### Presentation and clarity (W2, W4, and Q4)
> > > >**W2.** The abstract and introduction emphasize theory as part of the contribution, but the main text does not explicitly present any novel theoretical results.
> > > >
> > >
> > > Thank you for the suggestion. To address this, we have moved the most important theoretical contributions from the Appendix to the main text (please see Sec 3.3).
> > > More specifically, we have added the following formal statements to the main paper (which we will update fully before the discussion period ends).
> > >
> > > **Proposition 2 (Expressiveness of EGONETGNN)**. EGONETGNN is expressive enough to capture all dyad, open triad, closed triad, and open tetrad causal network motifs.
> > >
> > > We sketch the proof by dividing the statement into following two claims. The details of the proofs of these claims are in Appendix A.4.
> > >
> > > *Claim 1. EGONETGNN is as expressive as standard MPGNN in capturing dyad, open triad, and open tetrad causal network motifs.*
> > >
> > > *Claim 2. EGONETGNN also captures closed triad causal network motifs.*
> > >
> > > > **W4.** The paper only validates the model on synthetic and semi-synthetic datasets, without real-world intervention data.
> > > >
> > >
> > > As far as we know, there are no publicly available real-world interventional network datasets. With the exception of Yuan et al. 2021 who had access to a proprietary dataset, all our SOTA baselines evaluate their algorithms on synthetic and semi-synthetic benchmark datasets.
> > >
> > > > **Q4.** What are the key differences and advantages of EGONETGNN compared to recent methods such as AEMNet (ICASSP 2025) and CauGramer (ICLR 2025)? Could the framework potentially be extended to integrate attention or transformer-based mechanisms?
> > >
> > > As discussed in the related work (A.2) and the introduction, these works (AEMNet, CauGramer) use off-the-shelf message
> > > passing GNNs (e.g., GCN and GIN) and lack expressiveness to capture mechanisms involving local
> > > neighborhood structure. Yes, our method can be extended to integrate attention or transformer-based mechanisms in either feature mapping learning or exposure mapping learning by replacing/modifying MPGNN architectures. We will add this to the discussion section.

---

### Official Review · Reviewer_gdhX · 2025-10-31

**Soundness:** 2
**Presentation:** 2
**Contribution:** 2
**Rating:** 4
**Confidence:** 4

**Summary:**

This paper proposes a new method for estimating peer effects in the presence of interference. Instead of relying on assumptions about the exposure mapping function, their method is able to learn this function from data. The authors show the performance of their method on a variety of network datasets.

**Strengths:**

- The authors propose a novel method for an important and interesting problem.
- The authors extensively validate their method on synthetic data and a wide variety of different exposure mapping functions

**Weaknesses:**

In Section 3.2 the authors explain TARNet and CFR and how they apply balancing to their objective. As I understand it, in the original paper [1] the architecture of these two methods is the same; the difference is that CFR uses representation balancing with an IPM metric. In section 3.2, however, it seems like these two methods have a different architecture. It would be helpful if the authors could clarify this, and maybe also change the confusing naming of their two variants.

Related to this, the authors write that IPM balances the distribution $P(c,p|t)$. However, I am not sure I understand why this would be desirable.  In CFR, the representations are being balanced with respect to the treatment of interest. Here, given that the goal is to estimate peer effects, the representations are being balanced to a variable that is not the exposure of interest (which would be $\rho$). Could the authors give some more explanation as to why balancing with respect to the individual treatment would give better peer effect estimates?

It is unclear how the authors select the hyperparameters, especially the loss weights $\lambda$. Hyperparameter selection is a very important problem in treatment effect estimation because you cannot optimize them based on the loss of interest (which contains unobservable counterfactuals) [2]. Given that hyperparameter selection can have a very large impact on results, I would like the authors to discuss this in the paper.

On line 361 the authors write that they report PEHE with respect to peer effect estimation for 5 different simulations. What exactly is meant by simulations in this case? Does this mean that the data is always different in each simulation? Would it make more sense to use the same DGP, but with different random initializations of the model?

[1] Shalit, U., Johansson, F. D., & Sontag, D. (2017, July). Estimating individual treatment effect: generalization bounds and algorithms. In International conference on machine learning (pp. 3076-3085). PMLR.

[2] Curth, A., & Van Der Schaar, M. (2023, July). In search of insights, not magic bullets: Towards demystification of the model selection dilemma in heterogeneous treatment effect estimation. In International conference on machine learning (pp. 6623-6642). PMLR.

**Questions:**

Why do you only evaluate based on peer effect estimation performance? If I understand the method correctly, I would think that you could also learn direct and total effects?

On line 260-261: "Masked weights promotes representation that is invariant to irrelevant contexts ..." Why do masked weights promote this?

Could you explain why the two aggregations for $\rho$ are enough and relevant for all different exposure mapping functions? Are there any exposure mappings that the method cannot learn?

How well does your method perform compared to the baselines when the baselines make the right assumption about the exposure mapping function? I think this would provide additional insights into the performance of the different methods.

---

> ### Author Response · Authors · 2025-11-21
> **Rebuttal Part 1**
>
> Thank you for your detailed comments. We have provided point-wise responses below.
>
> >**W1**. In Section 3.2 the authors explain TARNet and CFR and how they apply balancing to their objective. As I understand it, in the original paper [1] the architecture of these two methods is the same; the difference is that CFR uses representation balancing with an IPM metric. In section 3.2, however, it seems like these two methods have a different architecture. It would be helpful if the authors could clarify this, and maybe also change the confusing naming of their two variants.
>
> This is a good observation. Yes, the original paper [1] uses CFR architecture similar to TARNet, without an autoencoder. Our EGONETGNN-CFR architecture implements an autoencoder architecture with reconstruction loss in addition to IPM balance loss because it helps to mitigate the potential loss in expressiveness while balancing the representations across treatment groups. We called this variant EGONETGNN-CFR because our idea is inspired by Assumption 1 in the original paper that uses an invertible representation assumption $\psi(\phi(x)) = x$ for all $x \in \mathcal{X}$, where $x$ is the input and $\phi$ is the feature mapping function. This is similar to the autoencoder architecture with $h=\phi(x)$ as the encoder layer, $\hat{x} = \psi(h)$ as the decoder layer, and objective to minimize $(x - \hat{x})^2$. To clarify the confusion, we will rename our variant that uses CFR with autoencoder as EGONETGNN-CFR+.
> We ran an additional experiment (Table R.1 below) with the original CFR without autoencoder to be included in the Appendix. The results show our variant of CFR with autoencoder has more robust performance across influence mechanisms than the original variant without autoencoder. For TARNet, the autoencoder would be a computational overhead and, thus, we stick with the original implementation.
>
>
> **Table R.1: Performance (in BlogCatalog Data in terms of PEHE) of three variants of outcome models in our method: EgoNetGNN-TARNet (without balancing), EgoNetGNN-CFR (original CFR with balancing), and EgoNetGNN-CFR+ (CFR w/ autoencoder). Experiments are conducted for five different data simulations, each with three random model initializations.**
>
>
> | Mechanism | EgoNetGNN-TARNet | EgoNetGNN-CFR+ (Our CFR w/ autoencoder) | EgoNetGNN-CFR (Original) | Best Baseline (INE-TARNet) |
> | -------- | -------- | -------- |-------- |-------- |
>  Clustering Coefficient | 2.13±1.88          | **0.95±0.54**        | *2.03±1.56*       | 2.35±0.71    |
> | Structural Diversity   | **1.47±0.90**          | *1.50±0.68*        | 1.57±0.67       | 4.78±1.09    |
> | Mutual Connections     | 2.86±1.31          | **2.24±1.55**        | 4.32±2.32       | *2.50±0.92*    |
> | Attribute Similarity   | 3.95±2.66          | *3.65±2.40*        | 4.14±1.84       | **3.59±1.83**    |
>
> >**W2.** Related to this, the authors write that IPM balances the distribution $P(c, p|t)$. However, I am not sure I understand why this would be desirable. In CFR, the representations are being balanced with respect to the treatment of interest. Here, given that the goal is to estimate peer effects, the representations are being balanced to a variable that is not the exposure of interest (which would be
> $\rho$). Could the authors give some more explanation as to why balancing with respect to the individual treatment would give better peer effect estimates?
>
> As discussed in Section 3.2 of the paper, $P(c,p|t)=P(p|t)P(c|p,t)$. In network settings, we need to balance the (covariate) representation for the entire exposure, including unit treatment and peer exposure, which is captured by the second term. The first term balances peer exposure representation for different unit treatment values. This balancing approach is straightforward extension to the original approach requiring low computation overhead and works for all network effects (direct, peer, and total). We will add this clarification in the paper.

---

> ### Author Response · Authors · 2025-11-21
> **Rebuttal Part 2**
>
> >**W3.** It is unclear how the authors select the hyperparameters, especially the loss weights $\lambda$. Hyperparameter selection is a very important problem in treatment effect estimation because you cannot optimize them based on the loss of interest (which contains unobservable counterfactuals) [2]. Given that hyperparameter selection can have a very large impact on results, I would like the authors to discuss this in the paper.
>
> We have used $\lambda_{ent}, \lambda_{sp}, \lambda_{bal}, \lambda_{cov}, \lambda_{L1}$ as hyperparameters in our loss function. For a set of hyperparameters, we choose reasonable values and for the rest we use Python's "Ray Tune" framework for hyperparameter tuning. Although ground truth causal effects are unavailable to truly tune hyperparameters, our error analysis (extended from Shalit et al. (2021)'s work) shows that error in factual outcome prediction (and IPM distance metric) can be used as a proxy for hyperparameter tuning.
>
> First, we describe the choice of values for regularization hyperparameters $\lambda_{ent}, \lambda_{sp},$ and $\lambda_{L1}$.
>
> - Entropy regularization coefficeint $\lambda_{ent}=1$ to promote mask weights $\mathbf{W}_{mask} \in [0,1]$ approaching 0 or 1 such that average entropy is low. An extremely low value does not enable the intended behavior of soft switch for enabling or disabling certain features and a large value could interfere with other loss terms.
> - Sparsity regularization coefficient $\lambda_{sp}=0.1$ to encourage sparse mask weights i.e., a few weights approaching 1. In conjuction with entropy loss, a value that is too high could lead all weights toward zero and a value that is too low could produce non-sparse weights.
> - L1-regularization coefficient $\lambda_{L1}=1$ for low-dimensional synthetic data to encourage highly sparse model parameters and $\lambda_{L1}=0.1$ for comparatively higher-dimensional semi-synthetic data to encourage sparse model parameters.
>
> We tune the coverage parameter $\lambda_{cov} \in \{0.01, 0.1\}$ for semi-synthetic data but choose a conservative $\lambda_{col}=0.01$ for synthetic data for effciency. For hyperparameter tuning/model selection, we vary learning rates along with other hyperparameters and, using 20% data for validation, select the model with lowest prediction loss for synthetic data and lowest sum of prediction and coverage loss for semi-synthetic data. We choose the covariate-balancing hyperparameter $\lambda_{bal}=0.8$ based on the analysis of the original paper (Shalit et al., 2017). We analyze the sensitivity of two important hyperparameters $\lambda_{bal}$ (new experiment in below) and $\lambda_{cov}$ in the Appendix (Table 6 in paper), and will add the above details in the paper. We will add the analysis for other mechanisms before the end of the discussion period.
>
> **Table R.2: Sensitivity analysis of hyperparameter $\lambda_{bal}$ in BlogCatalog Data with attribute similarity as underlying peer influence mechanism.**
> |         $\lambda_{bal}$             | 0.2       | 0.4       | 0.6       | 0.8       | 1.0       |
> |:-------------------|:----------|:----------|:----------|:----------|:----------|
> | Attribute Similarity | ++3.75±2.55++ | **3.43±2.29** | 3.94±2.64 | 3.78±2.55 | 4.33±3.54 |
>
>
> >**Q1.** Why do you only evaluate based on peer effect estimation performance? If I understand the method correctly, I would think that you could also learn direct and total effects?
>
> Indeed, the method can be generalized to direct and total effects as well. Because learning reliable exposure mapping functions is most pronounced in peer effect estimation and to reduce the scope (given this is a conference paper), we focus on peer effect estimation here and leave the other two to an extended version of the paper.
>
> >**Q2.** On line 260-261: "Masked weights promotes representation that is invariant to irrelevant contexts ..." Why do masked weights promote this?
>
> We use masked weights along with losses to promote for sparse and low entropy weights, i.e., only a few weights approaching 1 while others approaching 0. This helps the model to give importance to features/signals that actually matter in the underlying mechanism (captured by the outcome prediction) and ignore others, making the outcome prediction model more robust to irrelevant features.

---

> ### Author Response · Authors · 2025-11-21
> **Rebuttal Part 3**
>
> >**W4.** On line 361 the authors write that they report PEHE with respect to peer effect estimation for 5 different simulations. What exactly is meant by simulations in this case? Does this mean that the data is always different in each simulation? Would it make more sense to use the same DGP, but with different random initializations of the model?
>
>
> In our current experiments, the data is different in each simulation. We use five different arbitrary seeds for the same data generating process (DGP) or mechanism and refer to them as simulations. In our case, the models are designed to have deterministic/reproducible outputs for given data by fixing initializations and using deterministic operations.
>     However, one could use different random initializations of the model for the same DGP simulation to see the variance due to model initializations. As you have suggested, we have added an experiment (Table R.3 below) to study the variation in performance of our models and prominent baselines (for BlogCatalog data where true peer exposure depends on mutual connections). As one can see, even though in some simulations, explicit counting of graph motifs performs better, across simulations EgoNetGNN-CFR+ has the most robust performance.
>
> **Table R.3: Performance (in BlogCatalog Data in terms of PEHE) of our models and prominent baselines, with true peer exposure mechanism depending on mutual connections, for each simulation (i.e., DGP seed) and three random model initializations.**
>
> |   Simulation | EgoNetGNN-TARNet   | EgoNetGNN-CFR+   | GNN-MOTIFS   | INE_TARNet   | CauGramer   |
> |-------:|:-------------------|:-----------------|:--------------------|:-------------|:------------|
> |      0 | 2.58±0.46          | **2.23±0.55**        | 4.65±1.04           | *2.41±0.34*    | 5.60±2.83   |
> |      1 | *1.63±1.27*          | **0.72±0.31**        | 2.03±0.12           | 1.98±0.06    | 3.83±0.05   |
> |      2 | 3.45±0.91          | 4.06±1.21        | **3.07±0.53**           | *3.16±0.23*    | 5.38±2.62   |
> |      3 | *3.49±0.46*          | 3.51±0.78        | **3.04±0.16**          | 3.68±0.49    | 5.85±2.22   |
> |      4 | 3.16±2.36          | **0.70±0.10**        | 1.26±0.07           | *1.25±0.03*    | 5.25±1.82   |
> | Overall | 2.86±1.31          | **2.24±1.55**        | 2.81±1.26           | *2.50±0.92*    | 5.18±1.96   |
>
>
> >**Q4.** How well does your method perform compared to the baselines when the baselines make the right assumption about the exposure mapping function? I think this would provide additional insights into the performance of the different methods.
>
> Not all baselines are capable of representing the true mechanisms. Some baselines (NetEst and TNet) can make the right assumption about the exposure mapping function only when there is simple homogeneous influence between units. Other baselines (DWR, INE-TARNet, AEMNet, CauGramer) are capable of capturing exposure mapping functions that depend on attribute similarity. The more complex ones, GNN_TARNet-MOTIFS and INE-TARNet, can handle exposure mapping function that depends on number of mutual connections and GNN_TARNet-MOTIFS is also expressive enough to capture mechanisms involving clustering coefficient among treated but cannot handle attribute similarity. We ran additional experiments (Table R.5 below) which show the performance of our methods and baselines in the BlogCatalog Dataset when the true peer exposure mechanism depends on the fraction of treated peers. This is the simplest setting in which all baselines make the right exposure mapping function assumption.When the baselines know what the true mechanism is, our method performs better than or competitive to the baselines. Interestingly, even though the peer exposure mechanism is simple, the baselines suffer likely due to the fact that the data generation includes effect modifications and confounding.
>
> **Table R.4: Performance in BlogCatalog Dataset in terms of PEHE metric when true peer exposure mechanism depends on the fraction of treated peers. (Five data simulations with one model initialization)**
> |          Mechanism            | EgoNetGNN-TARNet   | EgoNetGNN-CFR+   | GNN_TARNet_MOTIFS   | INE_TARNet   | 1GNN_HSIC   | DWR-5      | AEMNet-CFR   | TNet       | NetEstimator   | CauGramer   |
> |:---------------------|:-------------------|:-----------------|:--------------------|:-------------|:------------|:-----------|:-------------|:-----------|:---------------|:------------|
> | Homogeneous | *2.40±2.74*          | **2.35±1.82**        | 4.25±2.22           | 4.00±1.75    | 13.90±2.80  | 16.65±1.96 | 14.36±3.81   | 13.49±2.94 | 11.76±2.20     | 8.48±6.17   |

---

> ### Comment · Reviewer_gdhX · 2025-11-21
> **Response Part 1**
>
> Thank you for your clarification! I appreciate that you changed the name to CFR+.
>
> Related to **W2**, I'm not sure I understand. My interpretation of balancing the covariate representation for the entire exposure means that
> $$P(c \mid p,t) = P(c \mid p',t') \quad \forall p,p',t,t'. \qquad \text{(1)}$$
> However, the balancing enforced is
> $$P(p,c \mid t) = P(p,c \mid t') \quad \forall t,t'. \qquad \text{(2)}$$
> Therefore, it follows that,
> $$P(p\mid t)P(c \mid p,t) = P(p\mid t')P(c \mid p,t') \quad \forall t,t'.$$
> However, this does not ensure balancing with respect to the entire exposure. As a counterexample, consider the following distributions of $t$, $p$, and $c$.
>
> $$
> P(t=0)=P(t=1)=\tfrac{1}{2}.
> $$
> $p \sim \mathrm{Uniform}(0,1)$ independently of $t$.
>
> $$
> c = p + \varepsilon,
> \quad
> \varepsilon \sim \mathrm{Uniform}(-0.1,0.1),
> $$
> with $\varepsilon$ independent of $\(p,t\)$.
>
> Then $\(c,p\)$ is independent of $\(t\)$, and therefore
> $$
> P(c,p \mid t=1) = P(c,p \mid t=0),
> $$
> which is the balancing objective (2).
>
> However, the claim that the covariates are balanced for the entire exposure (1) does not hold.
> $$
> P(c \mid p,t) = P(c \mid p)
>     = \mathrm{Uniform}(p-0.1,\; p+0.1),
> $$
> which depends on the value of $p$.
>
> Could you explain whether this reasoning is correct? If not, could you further explain what you mean?

---

> > ### Comment · Reviewer_gdhX · 2025-11-21
> > **Response Part 2+3**
> >
> > I appreciate the description of how the hyperparameters were chosen and the sensitivity analysis. I want to thank the authors for their efforts.
> >
> > **W4**:Thank you for your explanation about how the standard deviation is calculated and the additional experiments!
> >
> > I still have a couple of questions regarding the results. The reason I originally asked about whether you use different DGPs for the calculation of the standard deviation is that the standard deviation seems to be quite large in some experiments. It is important to know whether this is because performance is sensitive to the model initialization or because you always use different weights in the DGP.
> > From the results in Table R.3 the former does not seem to be the case.
> >
> > To ensure that we are talking about the same thing, I will use a very simplified example. How I understand it is that if your DGP were $Y = AX + BT$, you would sample $A$ and $B$ again in each "simulation"? If so, is this standard practice in treatment effect estimation literature? In my understanding, it seems that aggregating over different datasets does not provide a completely accurate image of the performance differences between the methods. Some simulations will naturally have a larger/smaller PEHE, which will lead to a high standard deviation that does not necessarily reflect how "stable" a method is.
> >
> > If I misunderstood and you do not sample $A$ and $B$ again, (1) What do you mean then by different simulations? (2) How would you explain the relatively large differences in performance for different simulations?
> >
> > **Q4**: Thank you for the extra experiments and analysis. This is indeed quite a surprising result. Is there some way to isolate the effect of exposure mapping learning from effect modification? Now, given that the performance differences with and without the correct exposure mapping specification are the same, it seems that the superior PEHE performance of EgoNetGNN with unknown exposure mappings is largely due to the fact that it can handle this effect modification. It would be interesting if you could provide some further intuition or experimental analysis for this.
> >
> > Again, I want to thank the authors for the effort they put into their rebuttal.

---

> > > ### Author Response · Authors · 2025-12-01
> > > **Responses to Follow-up on W4 and Q4**
> > >
> > > > Response to follow-up on W4
> > >
> > > Indeed, the performance of models is less sensitive to model initializations and more sensitive to data for the peer effect estimation setting. In your simplified example $Y = AX + BT$, assuming $Y$ as outcome, $X$ as confounders, and $T$ as treatment, for each new simulation, in addition to sampling $A$ and $B$, $T$ would also be sampled/derived from $T \sim Bernoulli(\sigma(WX))$. In the case of peer effects, we aim to demonstrate robustness across various possible configurations/patterns of peer treatment assignments or peer exposure conditions. To achieve this, we conduct multiple simulations, considering factual (observed/assigned) versus counterfactual (flipped) treatment assignments for the evaluation of peer effect estimation. The setup of considering multiple data simulations is common in causal inference (e.g., Arbour et al., KDD 2016), where significant variation arises from the fact that some patterns of neighborhood (treatment) assignments can be easily captured by the models, while others cannot. One standard approach to dealing with such large variance is to conduct multiple simulations and use distribution plots (e.g., violin plots) or the interquartile range to make the evaluation less sensitive to the outliers (due to difficult data or model convergence issues) (e.g., Arbour et al., KDD 2016).
> > >
> > > > Response to follow-up on Q4
> > >
> > > Our research question 1 (RQ1) examines the setting without complex effect modification (i.e., effect modification solely due to the unit's own treatment) in synthetic networks, and we demonstrate consistently better performance of our method. For semi-synthetic data with real-world network topology and attributes, we have considered a more challenging setting with complex effect modification depending on different contexts. There are several reasons the baseline models could fail in such a complex setting, and we investigate potential reasons (although this was not originally in the scope of our work). We note that there is always an issue with model convergence and difficult data in such experiments/analyses.
> > >
> > > First, we consider a setup with homogeneous exposure as true peer exposure mechanism, with and without effect modifications (EM).
> > > |                      | EgoNetGNN-TARNet   | EgoNetGNN-CFR+   | GNN-MOTIFS   | INE_TARNet   | 1GNN_HSIC   | DWR-5      | AEMNet-CFR   | TNet       | NetEstimator   | CauGramer   |
> > > |:---------------------|:-------------------|:-----------------|:--------------------|:-------------|:------------|:-----------|:-------------|:-----------|:---------------|:------------|
> > > | w/ EM | 2.40±2.74          | **2.35±1.82**        | 4.25±2.22           | 4.00±1.75    | 13.90±2.80  | 16.65±1.96 | 14.36±3.81   | 13.49±2.94 | 11.76±2.20     | 8.48±6.17   |
> > > | w/o EM |**1.05±0.93**          | 3.18±3.62        | 2.17±2.70           | 2.47±2.34    | 12.80±4.37  | 15.31±2.41 | 11.82±1.82   | 12.58±4.12 | 10.58±1.34     | 4.99±1.54   |
> > >
> > > Most approaches perform better without effect modification. For some methods, model convergence and variance due to irrelevant features could still affect mean performance. The next complexity is the setting of observed factual versus flipped counterfactual treatment setup. Most baselines are designed/evaluated for the setup with only one counterfactual setup none-treated (all-zero), and they are not versatile to handle multiple counterfactual patterns that flip treatments setting tests. In such simplified setting with single fixed counterfactual, EgoNetGNN-TARNet still has the best performance, with TNet, 1GNN_HSIC, and DWR-5 having competitive performances.
> > >
> > > |                      | EgoNetGNN-TARNet   | EgoNetGNN-CFR+   | GNN-MOTIFS   | INE_TARNet   | 1GNN_HSIC   | DWR-5     | AEMNet-CFR   | TNet      | NetEstimator   | CauGramer   |
> > > |:---------------------|:-------------------|:-----------------|:--------------------|:-------------|:------------|:----------|:-------------|:----------|:---------------|:------------|
> > > | w/o EM & w/o flip | 0.74±0.82          | 1.90±1.90        | 2.22±2.53           | 2.07±1.66    | 1.36±1.22   | 1.54±2.03 | 3.39±2.49    | 0.83±1.26 | 2.97±1.92      | 2.96±0.88   |

---

> > ### Author Response · Authors · 2025-12-01
> > **Response to Followup on W2**
> >
> > We thank the reviewer for clarifying the question with great detail and for the clever example. Indeed, balancing the covariate representation for the entire exposure means ensuring a similar distribution of covariates conditioned on any values of unit treatment and exposure conditions (Equation 1). Ideally, we should enforce such a balancing for pseudo-randomization across different peer exposure conditions. However, explicitly enforcing IPM balancing (using approaches such as the Wasserstein distance) for Equation 1 is non-trivial at best and computationally infeasible at worst. The challenges arise primarily because the peer exposures (p versus p') in the context of Equation 1 are multi-dimensional continuous values. Additionally, there may be several exposure conditions with mostly limited samples, which complicates the calculation of the Wasserstein distance (optimal transport) and may render it difficult or unreliable, even when using modern methods.
> > Our balancing technique is a computation-friendly and useful heuristic approach (as evidenced by the experiments), where for each stratum of peer exposure condition, we want balanced covariates across different treatment groups. As the reviewer pointed out, it does not perform the ideal balancing scenario in Eq. 1. However, even in network experiments where treatments are randomized, such balancing is not possible because peer exposure conditions depend not only on peer treatments but also on other covariates, therefore the randomization for peer exposures is not achieved. We instead control for different possible network contexts (derived from the observational attributed network), similar to regression adjustment, to deal with imbalance. We will clarify the ideal conditions for balancing, our heuristic approach, and the limitations in the updated PDF of the paper.

---

### Author Response · Authors · 2025-12-04
**Summary of changes**

Dear AC,

We have updated our paper with the following main changes, as per the feedback from reviewers and subsequent discussion. New changes are highlighted in blue text.

1) **Theoretical results in the main paper:** The formal statement for Proposition 2, along with a high-level proof sketch, is moved to the main paper. This addresses W2 by the reviewer EFAX and Q3.1 by the reviewer KRe6.

2) **Clarification on our variant of CFR with autoencoder and IPM balancing:** To address W1 and W2 by the Reviewer gdhX, including the follow-up discussion, we add clarification in Section 3.2 (page 6). We also include the experimental results in Table R.1, which are discussed in the rebuttal, in the Appendix under Research Question 3 (RQ3).

3) **Experiments with 5 data simulations with 3 random model initializations:** To address W4 by the reviewer gdhX, we update semi-synthetic experiments (e.g., Table 1 and Table 4) to include 5 data simulations with 3 random model initializations each. We also include the experimental results in Table R.3, discussed in the rebuttal, in the Appendix under research question (RQ2).

4) **Description of hyperparameters:** To address W3 by the reviewer gdhX and W6 by the reviewer KRe6, we include the description of missing hyperparameters in the Appendix. We also add the experiment on sensitivity to hyperparameter $\lambda_{bal}$ (Table R.2 in rebuttal) under RQ3.

5) **Research Question 5:** We include a new research question to study how models perform when the assumption regarding exposure mapping is satisfied (i.e., under homogeneous exposure) or when the model assumptions are violated (multi-hop confounding/interference, noisy, and missing features) according to the feedback from reviewers (Q4 by the reviewer gdhX; W1, Q1, and Q3 by the reviewer EFAX; W1, Q1, and Q2 by the reviewer KRe6; and W2 and Q2 by the reviewer DGdQ).

6) **Experiments on runtime and space:** We include the experimental results (Table R.6 in rebuttal) in the Appendix to supplement theoretical runtime analysis according to feedback from reviewers (W5 and Q2 by the reviewer EFAX and W5 by the reviewer KRe6).

7) **Other discussions, experiments, and clarifications:** We also include other important discussions, experiments, and clarifications (in addition to the six points above) presented during the rebuttal in the paper. These include the discussions about potential limitations and future research directions or extensions.

Thank you.
Authors

---

### Meta-Review · Area_Chair_HTMb · 2026-01-05

**Summary:**

This paper addresses the important problem of heterogeneous peer effect estimation under network interference when the true exposure mapping function is unknown or misspecified. Instead of relying on manually designed exposure mappings, the authors propose EGONETGNN, a graph neural network framework that automatically learns peer exposure representations from ego-net structures, node attributes, and edge features. The method is designed to be expressive enough to capture a wide range of causal network motifs and peer influence mechanisms. The paper combines architectural innovation with extensive synthetic and semi-synthetic evaluations, demonstrating robustness across diverse exposure mechanisms, confounding structures, and noisy or missing data scenarios.

**Reviewer Concerns:**

Reviewer gdhX focused on conceptual clarity and experimental methodology, questioning the correctness of representation balancing with respect to peer exposure, the justification for balancing on individual treatment, the lack of clarity in hyperparameter selection, and the interpretation of simulation variance across multiple data-generating processes.
Reviewer EFAX raised concerns about the reliance on strong causal assumptions such as unconfoundedness and neighborhood interference, the lack of explicit theoretical results despite claims of theory contribution, scalability to large graphs, absence of real-world interventional data, and incomplete runtime comparisons with baselines. The reviewer also asked about robustness to noisy or missing node features.
Reviewer KRe6 emphasized the lack of formal causal guarantees, identifiability, and consistency results, questioning whether reduced PEHE reflects causal correctness or merely representational expressiveness. The reviewer also raised concerns about interpretability of learned exposure embeddings, scalability, statistical reliability, incomplete baseline comparisons, and potential endogeneity from joint learning of exposure and outcome models.
Reviewer DGdQ questioned the practical relevance of the chosen exposure mechanisms, asked for justification grounded in empirical applications, and requested comparisons in settings where baselines make correct exposure assumptions. The reviewer also asked about applicability when graph information is incomplete.

**Reviewer Scores:**

I think Reviewer gdhX, KRe6, and DGdQ would not change their score. But for Reviewer EFAX, the question “What would happen if the unconfoundedness or neighborhood-interference assumptions (Assumptions 2–3) are violated?” can be addressed with new sensitivity experiments involving confounding and two-hop interference.

---

### Decision · Program_Chairs · 2026-01-26

Accept (Poster)